# The Regime of Rural Ecotourism Stakeholders in Poverty-Stricken Areas of China: Implications for Rural Revitalization

**DOI:** 10.3390/ijerph18189690

**Published:** 2021-09-14

**Authors:** Keke Sun, Zeyu Xing, Xia Cao, Weijia Li

**Affiliations:** 1School of Economics and Management, Harbin Engineering University, Harbin 150001, China; allthetimesmile@163.com (K.S.); caoxiaheu@163.com (X.C.); skk1218@126.com (W.L.); 2School of Management, Zhejiang University of Technology, Hangzhou 310023, China

**Keywords:** rural ecotourism, rural revitalization, evolutionary game theory, poverty-stricken areas

## Abstract

The rural ecotourism system can be defined as a complex association of stakeholders. This system of rural ecotourism in relatively poor areas of China can influence rural revitalization strategies. The purpose of this study is to plan a rural ecotourism system among the tourism enterprises, local residents and government by using an evolutionary game theory. Based on the theoretical analysis, an evolution game model for the three stakeholders is developed and the evolution process of strategies is described by replicator dynamic equations. Then, a simulation method and case was used to analyze the stability of interactions among the stakeholders and determine an equilibrium solution in the finite rationality case. Finally, specific control strategies were proposed to suppress instability and an ideal evolutionarily stable strategy was obtained. This provides a theoretical basis for achieving a win-win situation among the three parties. The results of this study suggest appropriate roles for stakeholders in the rural ecotourism project that provide management implications for rural tourism activities, local economy and rural revitalization.

## 1. Introduction

Rural tourism has strong market advantages, powerful connection-making functions, emerging industrial vitality and huge driving effects [1,2]. On the one hand, it is essential to promote harmony and achieving international exchanges; on the other hand, it can drive the economic development of poor regions, increasing employment opportunities for the poor, and play an increasingly important role in the cause of poverty reduction in the world [3,4]. In the past, rural tourism has also played a huge role in China’s poverty alleviation and development, with remarkable results [5,6,7,8]: firstly, with the implementation and upgrading of the action plan for the transformation and upgrading of tourism infrastructure in the “three regions and three states” deep poverty areas and the “three regions and three states” tourism ring road, the tourism ring road has been implemented. Secondly, with the implementation of the action plan for upgrading tourism infrastructure in the “three regions and three states” and the tourism loop, rural tourism in the “three regions and three states” deep poverty areas has achieved significant results in poverty alleviation. On 18 May 2020, the Information Office of the State Council reported that as of 17 May, 780 counties had been declared out of poverty, leaving only 52 poor counties in seven provinces and regions [8,9]. The poor counties are a major achievement in the battle against poverty, and from the way the poor counties have been removed from poverty and its effectiveness, there is a lot of “tourism power” in this. Finally, in recent years, rural tourism has explored and developed practical and effective ways to help alleviate poverty, such as, Manaus in Brazil, Cancun Island in Mexico, Las Vegas in the United States, the Baleares Islands in Spain, the French Alps, Phuket in Thailand, Bali in Indonesia, the Sinai Peninsula in Egypt and Morocco in Africa, which were previously very backward places, have all become world-famous tourist destinations as a result of rural tourism development, with great social, economic and cultural improvement and development. In particular, as highlighted by other authors [10,11], the reinforcement of local food productions and short food supply chains might play a key role in rural ecotourism revitalization. In China, such as “scenic spots leading villages”, “capable people leading households”, “cooperatives+ farmers”, and “enterprises+ farmers”, which are important achievements in China’s poverty alleviation efforts. Although rural tourism has made a series of achievements and will enter the final year of the battle against poverty in 2020, there are still many problems that need to be solved in the next “rural revitalization era”. For example, many regions lack planning in rural tourism development, blindly developing tourism projects without in-depth study of local tourism resources and advantages [12,13], design similar tourism products without distinguishing characteristics [14,15], excessively pursue short-term economic gains without long-term planning [11,15,16,17], and establish “temporary rural tourism projects”. In some regions, in the process of rural tourism, there is an imbalance between the interests of the poor, the local government, tourism developers and tourists [12,18]. In other areas, there are poor infrastructure facilities and a lack of professional management personnel, which makes it difficult to attract tourists [19,20]. Therefore, to face these problems, the Chinese government needs to seize the policy opportunities, build on the existing foundation and experience, further improve the work of rural tourism to help alleviate poverty, consolidate the achievements of rural tourism, tell the story of rural tourism in China, and further advance towards the “*rural revitalization in 2035*”.

The development of rural ecotourism in poverty-stricken areas has become a key to achieving environmental protection, rural revitalization and sustainable development [10,11,21,22,23]. Rural ecotourism is an emerging type of tourism which gained attention in rural areas rich in ecological resources for the purpose of promoting the development of rural areas and protecting the rural ecological environment [24,25,26]. In the context of ecological civilization, the development of green rural tourism has considerable strategic importance, not only to improve the ecological environment but also to increase economic income for local residents, creating a beautiful and livable environment and promoting common prosperity [27]. In recent years, rural ecotourism has become more and more recognized by the public as a new mode of tourism, and the development of rural tourism has become an effective way to promote rural revitalization [4]. Compared with sightseeing spots and city tours, rural ecotourism uses the unique resources of the countryside to change tourism from passive visits into active participation, making people feel the simplicity and happiness of returning to the basics, which is increasingly recognized by the public [28].

However, there are problems in the development of rural ecotourism, such as homogenization of projects, sloppy management, difficulties in financing, shortage of talent and serious environmental pollution [29]. Due to the asymmetrical distribution of rural ecotourism resources, uneven economic development between urban and rural areas, towns and villages, villagers and local governments often develop “rural ecotourism” under the banner of ecological conservation to pursue their own interests and performance, while secretly trampling on the natural ecological environment of the countryside [30,31,32].There is no balance between pre-conservation development and post-conservation development, which may lead to further deterioration of the rural ecological environment [33,34,35]. In the process of developing and building rural ecotourism, tourism enterprises will adjust their strategies according to local government policies, but may sacrifice rural ecological environment and engage in false promotion of rural ecotourism in order to maximize short-term benefits [36,37,38]. The strategies of tourism enterprises largely determine the implementation strategies of other stakeholders [39,40]. However, existing research has surprisingly failed to focus on stakeholder conflicts of interest and whether stakeholders are authentic in implementing rural ecotourism, especially in relatively poor areas. Whether existing models of cooperation and incentive policies are feasible, how well the mechanisms work and what impact they have needs to be more fully demonstrated. Therefore, it is necessary to study the stakeholders’ in interaction and strategic choice in the development and construction of rural ecotourism in poverty-stricken areas [41,42,43]. In our study, we plan an ecotourism system and analyse the relationship of the tourism enterprises, local residents and government by using an evolutionary game theory. Based on the theoretical analysis, an evolution game model for the three stakeholders is developed and the evolution process of strategies is described by replicator dynamic equations [32,44]. Then, we use simulation method and case study to analyze the stability of interactions among the stakeholders and determine an equilibrium solution in the finite rationality case. Finally, specific control strategies are proposed to suppress instability and an ideal evolutionarily stable strategy is obtained. This provides a theoretical basis for achieving a win-win situation among the three parties. Our paper has two contributions that are not fully addressed in the previous literature. Firstly, we focus on the tourism enterprises, local residents and government on rural ecotourism in terms of evolutionary game, and analyze the regime mechanism of rural ecotourism among the three stakeholders in the poverty-stricken areas of China. Secondly, we use case study and simulation to analyze how the different strategies of the three parties affect the rural ecotourism project and implication for rural revitalization. Based on the findings of the study, this paper provides countermeasures and recommendations for tourism enterprises, government, and local residents.

## 2. Literature Review

Rural tourism has long been recognized in certain parts of Europe as an effective catalyst of rural socio-economic regeneration for over a hundred years [45]. Along with the development of rural tourism worldwide, rural tourism concept has many interpretations. Tourism activity in rural areas has remarkably increased in all the developed countries and developing countries worldwide, which has played a key role in the development of rural areas that were economically and socially depressed [26,46]. Rural tourism is a multi-stakeholder process that requires the joint efforts of all stakeholders [30,32,36,47,48]. The relevant studies on rural tourism include the following aspects. Firstly, study on the problems and countermeasures in rural tourism. In view of the problems in rural tourism, such as insufficient industrial characteristics, poor quality of residences and poor channels of return of benefits, there is a need to precisely characterise the rural tourism industry, invest in human resources for rural tourism and carefully design the return mechanism of tourism benefits [12,15,16]. In the process of rural tourism, problems such as the ‘tragedy of the commons’ can occur, which can hinder the development of rural tourism projects and require specific analysis [49]. In China, the following problems exist in rural tourism: weakness of high value-added industrial links, lack of impetus from core rural tourism enterprises, insufficient cooperation in rural tourism projects, single function of rural tourism industry, short industrial chains and insufficient localization of rural industrial chains [6,50]. Therefore, we should actively cultivate core rural tourism enterprises [51], strengthen the integration of the rural tourism industry chain [52], accelerate the localization of the rural tourism industry chain and strengthen regional cooperation in rural tourism [52,53]. The second is the study of rural tourism models and pathways. It is necessary to further explore the three major types of rural tourism models: top-down, bottom-up and top-down cooperation, and to propose rural tourism models [54,55], including the government-enterprise cooperation model, the strategic alliance model, the leisure agriculture and rural tourism model and the regional linkage model [56], the rural ecological agriculture model, the tourism+ featured town model, the Organization-Resource-Humanity-Benefit (O-RHB) model and the collaborative participation model of multiple subjects [57,58]. Thirdly, it is a study on the mechanism and effect of rural tourism. With the help of methods such as the Moran index and the spatial Durbin model, explore the spatial correlation characteristics and spatial spillover effects between the level of tourism development and poverty alleviation in each dimension from a multi-dimensional perspective, and conclude that tourism development has a significant effect on poverty alleviation in the economic, living and environmental dimensions [58]. Alternatively, the number of people living in poverty was used as a direct poverty measure, and a social accounting matrix was used to simulate the poverty alleviation effect of tourism, which was found to be an industrial tool for poverty alleviation in Ecuador [59]. Some scholars outside of the first instance have used a three-stage Data Envelopment Analysis (DEA) model to analyze the efficiency of rural tourism in different regions, thus providing valuable strategies and suggestions to promote the smooth implementation of tourism poverty alleviation [60,61]. Of course there are some who argue that rural tourism does not always bring positive poverty alleviation effects, but may also bring negative effects such as environmental changes and loss of traditional culture [62].

The term ecotourism is surrounded by confusion [63]. It has been defined as ‘‘responsible travel to natural areas that conserves the environment and sustains the well-being of local people [64]’’. However, it is contended here that, regardless of definition, ecotourism is an instigator of change. It is inevitable that the introduction of tourists to areas that were previously seldom visited by outsiders will place new demands upon the environment associated with new actors, new activities, and new facilities. This will involve the forging of new relationships between people and environment, between peoples with different life-styles, and between a wide variety of forces for both change and stability. These forces act at a diversity of scales from global to local. Change is desired by most of the players involved in ecotourism, many of whom would like to see what they regard as an improvement in the existing situation. Rural ecotourism is a kind of tourism that takes the protection of the natural ecological environment as the premise and relies on the good natural ecological environment and unique human resources in the countryside to carry out ecological experience, ecological education, ecological cognition, while obtaining physical and mental enjoyment [65]. Rural ecotourism is the main form of sustainable tourism [21,66]. Rural ecotourism takes the countryside as the backdrop and the distinctive characteristic environment as the main landscape [67]. It mainly refers to the tourism model of ecological education, ecological experience and ecological awareness based on the concept of sustainable development, ecological environmental protection, harmonious development of human and nature, and the goal of maintaining the coordinated economic, social and environmental development of rural areas, relying on a better natural ecological environment and unique human ecological system [27,68]. Currently, environmental pollution and damage in rural tourism areas are serious and all stakeholders are deeply affected by the environmental damage in rural tourism areas [31,69]. In order to develop sustainable rural tourism and promote rural ecological and environmental protection and economic development, the relationships and evolutionary processes of the main stakeholders should be mainly analyzed and studied [70,71,72]. As a method tool, evolutionary game theory has been applied to many fields [73,74,75,76,77,78]. In recent years, evolutionary game theory has been increasingly applied in the field of rural tourism [79,80] or ecotourism [32,81], with less attention paid to the poverty-stricken areas, especially in China. Previous studies have mainly considered the relationship between government, firms or residents and firms. However, few scholars have examined the rural ecotourism stakeholders in poverty-stricken areas, highlighting implications for rural revitalization. This paper constructs a tripartite evolutionary game model from the perspective of an evolutionary game combined with simulation and case study analysis to address the issues of local government regulation and resident participation in the implementation of rural ecotourism by tourism enterprises in the construction of rural ecotourism projects. The model is used to analyze the strategy combinations of the three stakeholders, and to provide a long-term stable implementation strategy for the three stakeholders in the rural ecotourism project.

## 3. Model Building and Analysis

### 3.1. Analysis of the Interest Issues of Each Stakeholder

There are three main game stakeholders in rural ecotourism: tourism enterprises involved in rural ecotourism, local residents and the government. The strategies that tourism enterprises can choose are substantive rural ecotourism and spurious rural ecotourism, the strategies that local residents can choose are active participation and negative participation, and the strategies that the government can choose are strict regulation and formal regulation. There are the following game relations among stakeholders:(1)The game between government and tourism enterprises

The government is the supporter, participant and supervisor of rural ecotourism. There are two strategies of government in the evolutionary game of rural ecotourism: strict regulation and formal regulation. Under strict regulation, the government will support tourism enterprises with policies and provide sufficient financial support, and reward and punish tourism enterprises for substantial and spurious behavior, respectively. The government will recover financial support for enterprises that spuriously conduct rural ecotourism. Conversely, if the government carries out formal regulation, it will be inactive in the process of rural ecotourism in most cases, and provide policy support but only less funding to tourism enterprises. Depending on the government’s different strategies, tourism enterprises will constantly adjust and thus make the appropriate strategic choices. 

(2)The game between government and local residents

The government with strict supervision vigorously publicizes policies related to rural ecotourism, thus encouraging local residents to actively participate in rural tourism projects and rewarding those who do so. The government with formal supervision, on the other hand, publicizes the relevant policies less vigorously and does not offer any rewards to local residents for their active participation. Local residents are guided by the government and choose their own strategies based on their understanding of the policies.

(3)The game between tourism enterprises and local residents

Local residents will take shares in rural tourism projects with assets such as land and houses, and work for tourism enterprises. The tourism enterprises that substantively conduct rural ecotourism will pay dividends to local residents at a certain interest rate, take on some local residents to work in their tourism projects, and pay them basic wages and bonuses. Conversely, tourism enterprises that spuriously conduct rural ecotourism rarely take on local residents, and if they do, they only give them a small bonus. Local residents may earn income from other business activities while participating in rural ecotourism projects. As a result, local residents make adjustments and strategic choices based on the income they receive and the company’s attitude toward them.

### 3.2. Basic Game Hypothesis and Model Construction

**Hypothesis** **1** **(H1).***The**stakeholders are tourism enterprises, local residents and the government, and all of them have limited rationality, aiming to maximize their own interests*.

**Hypothesis** **2** **(H2).***Local residents take equity in the tourism enterprise with assets such as land and houses, and the amount of equity is denoted by W. If the tourism enterprise adopts the strategy of “substantial rural ecotourism”, then the total input of the enterprise is F_1_. And when the local residents actively participate, the total return of the project is E_1_; when the local residents passively participate, the total return of the project is E_2_. If the tourism enterprise adopts the strategy of “**spuriously rural ecotourism”, then the total input of the enterprise is F_2_, and the total revenue of the project is E_3_**when the local residents actively participate, and the total revenue of the project is E_4_**when the local residents negatively participate. (F_2_ < F_1_)). The tourism enterprise that adopts substantial behavior will receive good comments from local residents and others, which will have a good impact on the enterprise’s image and enhance its social image, and these additional benefits will be recorded as* I.

**Hypothesis** **3** **(H3).**
*The tourism enterprise will pay dividends to the local residents at a certain interest rate on their share capital. When a tourism enterprise substantially conducts rural ecotourism, the tourism enterprise will pay dividends according to the interest rate of μ_1_*
*regardless of whether government regulation is in place. When a tourism enterprise*
*spuriously conducts rural ecotourism, the tourism enterprise will pay dividends according to the interest rate of μ_2_ if the government strictly regulates it, and if the government formally regulates it, then the tourism enterprise will K_2_ < K_1_) (to simplify the formula, the labor costs of local residents are not taken into account).*


**Hypothesis** **4** **(H4).***Negatively engaged residents have a passive and perfunctory attitude, are not enthusiastic about their work, and receive only a basic salary and no bonus from the company or the government. Negatively participating local residents will have more time to do other activities (such as selling agricultural products, making special handicrafts, etc.) and earn income, which is denoted by N*.

**Hypothesis** **5** **(H5).***The smooth implementation of rural ecotourism requires the government to increase the promotion and publicity of relevant policies, and also requires local residents to have a thorough understanding of the policies. The promotion and publicity of rural ecotourism policies by the government with strict regulation is greater than that by the government with formal regulation, C_1_**indicating the publicity cost of the government with strict regulation and C_2_ indicating the publicity cost of the government with formal regulation (C_2_ < C_1_). The local residents who actively participate in the policy will take the initiative to understand the policy.**Due to the different promotion and publicity efforts of the government and the different understanding ability of the local residents, the increased benefits of the local residents are different. α_1_ and α_2_ denote the publicity efforts of the government with strict regulation and formal regulation, respectively. β indicates the ability of local residents to understand the policy. The increase in benefits for local residents who actively participate under strict government regulation is indicated by α_1_βP, while the increase in benefits for local residents who actively participate under formal government regulation is indicated by α_2_βP. Negatively participating local residents do not actively seek to understand the policy and therefore do not increase their benefits*.

**Hypothesis** **6** **(H6).***In order to promote enterprises’ participation in rural ecotourism activities and reduce the risks and costs of enterprises’ participation in rural ecotourism, the local government will provide policy support and financial support to tourism enterprises. ΔF is the cost savings of tourism enterprises involved in rural ecotourism under government policy support. For example, the government facilitates companies involved in rural ecotourism projects through low interest rate loans or tax incentives. H_1_ is the financial support of government with strict regulation to tourism enterprises, i.e., government subsidies.**H_2_ is the financial support of government with formal regulation to tourism enterprises (**H_2_ < H_1_)*.

**Hypothesis** **7** **(H7).***C_3_ is the cost of monitoring of government with strict regulation, R is the praise reward given by the government to enterprises that substantially conduct rural ecotourism. The government will recover the financial subsidy and impose penalties on enterprises that adopt**spurious practices. R is the fine imposed by the government on tourism enterprises that adopt**spurious practices,**λ is the probability of being detected by the government when tourism enterprises**spuriously conduct rural ecotourism, and**M indicates the reward that the government will give to the residents who actively participate*.

**Hypothesis** **8** **(H8).**
*When the government “strictly regulates”, its credibility and self-image will be enhanced, which is recorded as T.*
*L is the increase in the welfare of the society as a whole after the tourism enterprise has carried out a substantial project, and*
*D indicates the loss in the overall income of the society due to the tourism enterprise’s*
*spurious rural ecotourism.*


According to the above assumptions, the payoff matrix of different strategy combinations for the three subjects can be obtained, as shown in Table 1. 

### 3.3. Analysis of Evolution Game Model of Rural Ecotourism

The probability of a tourism company choosing a substantial rural ecotourism strategy is *x*, and the probability of a tourism company choosing a spurious rural ecotourism strategy is (1 − *x*). The probability of local residents choosing an active participation strategy is *y*, and the probability of choosing a negative participation strategy is (1 − *y*). The probability of the government choosing a strict regulation strategy is *z*, and the probability of choosing a formal regulation strategy is (1 − *z*).

#### 3.3.1. Expected Return Function of Stakeholders in Rural Ecotourism

(1)Equilibrium analysis of tourism enterprises

The expected benefit for tourism enterprises choosing to substantially conduct rural ecotourism is:$$\begin{array}{l} U_{11}=yz\left(E_{1}+I-F_{1}-\mu_{1}W-S-K_{1}+\Delta F+H_{1}+R\right)+\left(1-y\right)z\left(E_{2}+I-F_{1}-\mu_{1}W-S+\Delta F+H_{1}+R\right)\\ +y\left(1-z\right)\left(E_{1}+I-F_{1}-\mu_{1}W-S-K_{1}+\Delta F+H_{2}\right)+\left(1-y\right)\left(1-z\right)\left(E_{2}+I-F_{1}-\mu_{1}W-S+\Delta F+H_{2}\right) \end{array}$$

The expected benefit of tourism enterprises choosing to spuriously conduct rural ecotourism is:$$\begin{array}{l} U_{12}=yz\left(E_{3}-F_{2}-\mu_{1}W-S-K_{2}+\Delta F+\left(1-\lambda \right)H_{1}-\lambda G\right)+\left(1-y\right)z\left(E_{4}-F_{2}-\mu_{1}W-S+\Delta F+\left(1-\lambda \right)H_{1}-\lambda G\right)\\ +y\left(1-z\right)\left(E_{3}-F_{2}-\mu_{2}W-S-K_{2}+\Delta F+H_{2}\right)+\left(1-y\right)\left(1-z\right)\left(E_{4}-F_{2}-\mu_{2}W-S+\Delta F+H_{2}\right) \end{array}$$

The average expected return of tourism enterprises is:$${\overset{-}{U}}_{1}=xU_{11}+\left(1-x\right)U_{12}$$

(2)Equilibrium analysis of local residents

The expected benefit of local residents choosing to actively participate is:$$\begin{array}{l} U_{21}=xz\left(\mu_{1}W+S+K_{1}+\alpha_{1}\beta P+M\right)+x\left(1-z\right)\left(\mu_{1}W+S+K_{1}+\alpha_{2}\beta P\right)\\ +\left(1-x\right)z\left(\mu_{1}W+S+K_{2}+\alpha_{1}\beta P+M\right)+\left(1-x\right)\left(1-z\right)\left(\mu_{2}W+S+K_{2}+\alpha_{2}\beta P\right) \end{array}$$

The expected benefit of local residents choosing to negatively participate is:$$U_{22}=xz\left(\mu_{1}W+S+N\right)+x\left(1-z\right)\left(\mu_{1}W+S+N\right)+\left(1-x\right)z\left(\mu_{1}W+S+N\right)+\left(1-x\right)\left(1-z\right)\left(\mu_{2}W+S+N\right)$$

The average expected return of local residents is:$${\overset{-}{U}}_{2}=yU_{21}+\left(1-y\right)U_{22}$$

(3)Equilibrium analysis of the government

The expected benefit for the government choosing strict regulation is:$$\begin{array}{l} U_{31}=xy\left(-C_{1}-H_{1}-C_{3}-R-M+T+L\right)+x\left(1-y\right)\left(-C_{1}-H_{1}-C_{3}-R+T+L\right)\\ +\left(1-x\right)y\left(-C_{1}-C_{3}-\left(1-\lambda \right)H_{1}+\lambda G-M+T-D\right)+\left(1-x\right)\left(1-y\right)\left(-C_{1}-C_{3}-\left(1-\lambda \right)H_{1}+\lambda G+T-D\right) \end{array}$$

The expected benefit of the government’s choice of form regulation is:$$U_{32}=xy\left(-C_{2}-H_{2}+L\right)+x\left(1-y\right)\left(-C_{2}-H_{2}+L\right)+\left(1-x\right)y\left(-C_{2}-H_{2}-D\right)+\left(1-x\right)\left(1-y\right)\left(-C_{2}-H_{2}-D\right)$$

The average expected return to the government is:$${\overset{-}{U}}_{3}=zU_{31}+\left(1-z\right)U_{32}$$

#### 3.3.2. Analysis of the Evolutionary Stabilization Strategy of Rural Ecotourism Stakeholders

(1)Analysis of the replication dynamics of tourism companies

The replication dynamic equation for tourism firms is:$$F(x)=x\left(U_{11}-{\overset{-}{U}}_{1}\right)=x\left(1-x\right)\left[y\left(E_{1}-E_{2}-E_{3}+E_{4}-K_{1}+K_{2}\right)+z\left(R+\lambda H_{1}+\lambda G+\mu_{1}W-\mu_{2}W\right)+\left(F_{2}-F_{1}+E_{2}-E_{4}+I-\mu_{1}W+\mu_{2}W\right)\right]$$

Analyze the evolutionary stabilization strategy of tourism companies:

①When $z=\frac{y\left(E_{2}+E_{3}-E_{1}-E_{4}+K_{1}-K_{2}\right)+\left(F_{1}-F_{2}+E_{4}-E_{2}-I+\mu_{1}W-\mu_{2}W\right)}{R+\lambda H_{1}+\lambda G+\mu_{1}W-\mu_{2}W}$, the equation F(x)=0 holds constant, which means that all levels are steady state, at which time the probability of strategy choice of tourism companies does not change over time. ②When $z\neq \frac{y\left(E_{2}+E_{3}-E_{1}-E_{4}+K_{1}-K_{2}\right)+\left(F_{1}-F_{2}+E_{4}-E_{2}-I+\mu_{1}W-\mu_{2}W\right)}{R+\lambda H_{1}+\lambda G+\mu_{1}W-\mu_{2}W}$, let F(x)=0, and x=0,x=1 are two stable points. Derivative of F(x) gives: $F^{\prime }\left(x\right)=\left(1-2x\right)\left[y\left(E_{1}-E_{2}-E_{3}+E_{4}-K_{1}+K_{2}\right)+z\left(R+\lambda H_{1}+\lambda G+\mu_{1}W-\mu_{2}W\right)+\left(F_{2}-F_{1}+E_{2}-E_{4}+I-\mu_{1}W+\mu_{2}W\right)\right]$ Due to $R+\lambda H_{1}+\lambda G+\mu_{1}W-\mu_{2}W>0$, two scenarios can be discussed. (i)When $E_{1}-E_{2}-E_{3}+E_{4}-K_{1}+K_{2}>0$ and $F_{2}-F_{1}+E_{2}-E_{4}+I-\mu_{1}W+\mu_{2}W>0$,$F^{\prime }\left(1\right)<0$ and $F^{\prime }\left(0\right)>0$. Then x=1 is a stabilization strategy, and tourism companies will choose substantial rural ecotourism.(ii)When the above conditions are not met, the following two cases are classified. When $z>\frac{y\left(E_{2}+E_{3}-E_{1}-E_{4}+K_{1}-K_{2}\right)+\left(F_{1}-F_{2}+E_{4}-E_{2}-I+\mu_{1}W-\mu_{2}W\right)}{R+\lambda H_{1}+\lambda G+\mu_{1}W-\mu_{2}W}$, it can be concluded that $F^{\prime }\left(1\right)<0$ and $F^{\prime }\left(0\right)>0$. Then x=1 is a stabilization strategy, and tourism companies will choose substantial rural ecotourism. When $z<\frac{y\left(E_{2}+E_{3}-E_{1}-E_{4}+K_{1}-K_{2}\right)+\left(F_{1}-F_{2}+E_{4}-E_{2}-I+\mu_{1}W-\mu_{2}W\right)}{R+\lambda H_{1}+\lambda G+\mu_{1}W-\mu_{2}W}$, it can be concluded that $F^{\prime }\left(0\right)<0$ and $F^{\prime }\left(1\right)>0$. Then x=0 is a stabilization strategy, and tourism companies will choose spurious rural ecotourism.

Solving the replicated dynamic equation yields the evolutionary process of the behavioral choice strategy of tourism firms, as shown in Figure 1.

(2)Analysis of the replication dynamics of local residents

The replication dynamic equation for local residents is:
$G(y)=y\left(U_{21}-{\overset{-}{U}}_{2}\right)=y\left(1-y\right)\left[x\left(K_{1}-K_{2}\right)+z\left(\alpha_{1}\beta P-\alpha_{2}\beta P+M\right)+K_{2}+\alpha_{2}\beta P-N\right]$.

Analyze the evolutionary stabilization strategies of the local residents.

①When $z=\frac{x\left(K_{2}-K_{1}\right)+N-K_{2}-\alpha_{2}\beta P}{\alpha_{1}\beta P-\alpha_{2}\beta P+M}$, the equation G(y)=0 holds constant, which means that all levels are steady state, at which time the probability of strategy choice of local residents does not change over time. ②When $z\neq \frac{x\left(K_{2}-K_{1}\right)+N-K_{2}-\alpha_{2}\beta P}{\alpha_{1}\beta P-\alpha_{2}\beta P+M}$, let G(y)=0, and y=0 and y=1 are two stable points. Derivative of G(y) gives: $G^{\prime }\left(y\right)=\left(1-2y\right)\left[x\left(K_{1}-K_{2}\right)+z\left(\alpha_{1}\beta P-\alpha_{2}\beta P+M\right)+K_{2}+\alpha_{2}\beta P-N\right]$.Due to $\alpha_{1}\beta P-\alpha_{2}\beta P+M>0$ and $K_{1}-K_{2}>0$, two scenarios can be discussed. (i)When $K_{2}+\alpha_{2}\beta P-N>0$, it can be concluded that $G^{\prime }\left(1\right)<0$ and $G^{\prime }\left(0\right)>0$. Then y=1 is a stabilization strategy, and local residents will choose active participation.(ii)When the above conditions are not met, the following two cases are classified. When $z>\frac{x\left(K_{2}-K_{1}\right)+N-K_{2}-\alpha_{2}\beta P}{\alpha_{1}\beta P-\alpha_{2}\beta P+M}$, it can be concluded that $G^{\prime }\left(1\right)<0$ and $G^{\prime }\left(0\right)>0$. Then y=1 is a stabilization strategy, and local residents will choose active participation. When $z<\frac{x\left(K_{2}-K_{1}\right)+N-K_{2}-\alpha_{2}\beta P}{\alpha_{1}\beta P-\alpha_{2}\beta P+M}$, it can be concluded that $G^{\prime }\left(0\right)<0$ and $G^{\prime }\left(1\right)>0$. Then y=0 is a stabilization strategy, and local residents will choose negative participation.

Solving the replicated dynamic equation yields the evolutionary process of the behavioral choice strategy of local residents, as shown in Figure 2.

(3)Analysis of the replication dynamics of the government

The replication dynamic equation for the government is:$$H(z)=z\left(U_{31}-{\overset{-}{U}}_{3}\right)=z\left(1-z\right)\left[x\left(-R-\lambda H_{1}-\lambda G\right)+y\left(-M\right)+C_{2}-C_{1}-C_{3}-\left(1-\lambda \right)H_{1}+H_{2}+\lambda G+T\right]$$

Analyze the evolutionary stabilization strategies of the government.

①When $x=\frac{-yM+C_{2}-C_{1}-C_{3}-\left(1-\lambda \right)H_{1}+H_{2}+\lambda G+T}{R+\lambda H_{1}+\lambda G}$, the equation H(z)=0 holds constant, which means that all levels are steady state, at which time the probability of strategy choice of the government does not change over time. ②When $x\neq \frac{-yM+C_{2}-C_{1}-C_{3}-\left(1-\lambda \right)H_{1}+H_{2}+\lambda G+T}{R+\lambda H_{1}+\lambda G}$, let H(z)=0, and z=0 an z=1 are two stable points. Derivative of H(z) gives: $H^{\prime }\left(z\right)=\left(1-2z\right)\left[-x\left(R+\lambda H_{1}+\lambda G\right)-yM+C_{2}-C_{1}-C_{3}-\left(1-\lambda \right)H_{1}+H_{2}+\lambda G+T\right]$.Due to $R+\lambda H_{1}+\lambda G>0$, two scenarios can be discussed.(i)When $C_{2}-C_{1}-C_{3}-\left(1-\lambda \right)H_{1}+H_{2}+\lambda G+T<0$, it can be concluded that $H^{\prime }\left(0\right)<0$ and $H^{\prime }\left(1\right)>0$. Then z=0 is a stabilization strategy, and the government will choose formal regulation.(ii)When the above conditions are not met, the following two cases are classified. When $x>\frac{-yM+C_{2}-C_{1}-C_{3}-\left(1-\lambda \right)H_{1}+H_{2}+\lambda G+T}{R+\lambda H_{1}+\lambda G}$, it can be concluded that $H^{\prime }\left(0\right)<0$ and $H^{\prime }\left(1\right)>0$. Then z=0 is a stabilization strategy, and the government will choose formal regulation. When $x<\frac{-yM+C_{2}-C_{1}-C_{3}-\left(1-\lambda \right)H_{1}+H_{2}+\lambda G+T}{R+\lambda H_{1}+\lambda G}$, it can be concluded that $H^{\prime }\left(1\right)<0$ and $H^{\prime }\left(0\right)>0$. Then z=1 is a stabilization strategy, and the government will choose strict regulation.

Solving the replicated dynamic equation yields the evolutionary process of the behavioral choice strategy of the government, as shown in Figure 3.

From the above evolutionary stability analysis, it can be seen that the evolutionary equilibrium of the group decision of tourism enterprises varies with the proportion y of local residents actively participating and the proportion *z* of the government with strict supervision in the evolutionary process. The evolutionary equilibrium of the group decision of local residents varies with the proportion *x* of tourism enterprises substantially conducting rural ecotourism and the proportion z of the government with strict supervision. The evolutionary equilibrium of the group decision of the government also varies with the proportion of tourism enterprises substantially conducting rural ecotourism and the proportion *y* of local residents actively participating in the evolutionary process. Since the values of *x*, *y* and *z* change over time with the evolutionary process, and the equilibrium state of the game system is not robust to small perturbations of *x*, *y* and *z*, it is not possible to promote the evolution of the tripartite game to the expected steady state by simply adjusting the initial conditions. In this paper, we aim to promote the evolution of the tripartite game to a socially rational model, i.e., to make the tripartite game evolve to an ideal decision state where tourism enterprises substantially conduct rural ecotourism, local residents actively participate, and the government strictly regulates (x=1, y=1, z=1). Therefore, by controlling or adjusting the relevant variables, the behavior of the participants can be guided to evolve in the desired direction.

(1) When $E_{1}-E_{2}-E_{3}+E_{4}-K_{1}+K_{2}>0$ and $F_{2}-F_{1}+E_{2}-E_{4}+I-\mu_{1}W+\mu_{2}W>0$, there is x→1. That is, tourism enterprises eventually tend to choose to substantially conduct rural ecotourism strategy. At this point, the smaller the values of (*K*_1_ − *K*_2_), (*F*_1_ − *F*_2_) and (*μ*_1_ − *μ*_2_), and the larger the value of *I*, the greater the probability that the above inequality holds will be. Therefore, when tourism companies substantially conduct rural ecotourism with less bonuses and dividends to local residents, the total investment in substantially conducting rural ecotourism is less, and the additional benefits such as the improvement of social image obtained due to substantially conducting rural ecotourism are higher, it is conducive to prompt tourism companies to choose the strategy of substantially conducting rural ecotourism. When *E*_1_ − *E*_2_ − *E*_3_ + *E*_4_ − *K*_1_ + *K*_2_ < 0 or *F*_2_ − *F*_1_ + *E*_2_ − *E*_4_ + *I*
*− μ*_1_*W* + *μ*_2_*W* < 0, then $z>\frac{y\left(E_{2}+E_{3}-E_{1}-E_{4}+K_{1}-K_{2}\right)+\left(F_{1}-F_{2}+E_{4}-E_{2}-I+\mu_{1}W-\mu_{2}W\right)}{R+\lambda H_{1}+\lambda G+\mu_{1}W-\mu_{2}W}$, there is x→1. At this point, the greater the values of R,λ and *G*, then the larger the denominator, the more likely the inequality will hold. Therefore, increasing the government’s praise and reward for companies that substantially conduct rural ecotourism, increasing the probability of being discovered by the government when tourism companies spuriously conduct rural ecotourism, and increasing the government’s fines for companies that spuriously conduct rural ecotourism will facilitate the evolution of tourism companies toward a strategy of choosing to substantially conduct rural ecotourism.

(2) When $K_{2}+\alpha_{2}\beta P-N>0$, there is y→1. That is, local residents eventually tend to choose to actively participate. Therefore, when tourism companies increase bonuses for residents who actively participate, the government increases the publicity of rural ecotourism policies, residents have a better understanding of rural ecotourism policies, and they earn less income from other activities while passively participating in rural tourism activities, it will help to motivate local residents to choose the active participation strategy. When $K_{2}+\alpha_{2}\beta P-N<0$, then $z>\frac{x\left(K_{2}-K_{1}\right)+N-K_{2}-\alpha_{2}\beta P}{\alpha_{1}\beta P-\alpha_{2}\beta P+M}$, there is y→1. At this point, the greater the value of M, the more likely the inequality will hold. Therefore, increasing the government’s incentives for residents who actively participate will facilitate the evolution of local residents toward choosing active participation strategy.

(3) When $x<\frac{-yM+C_{2}-C_{1}-C_{3}-\left(1-\lambda \right)H_{1}+H_{2}+\lambda G+T}{R+\lambda H_{1}+\lambda G}$, there is z→1. That is, the government eventually tends to choose to strictly regulate. At this point, the smaller the values of $\left(C_{1}-C_{2}\right)$, $C_{3}$, R and M, the greater the values of G and T, the more likely the inequality will hold. Therefore, if the government’s publicity cost is lower when strictly regulating, the cost of conducting supervision is lower, the rewards for companies that substantially conduct rural ecotourism and residents who actively participate are smaller, the punishment for companies that spuriously conduct rural ecotourism is higher, and the improvement of its own image due to strict regulation is higher, it will be favorable for the government to evolve toward the strategy of choosing strict regulation.

## 4. Case Analysis and Numerical Simulation

In order to better describe the impact of various parameters on the strategic choices of the government, tourism companies and local residents, we use a combination of case study and numerical simulations to quantitatively examine the impact of different parameters on the choice behavior of the three parties.

Steep Shahe Village, Huoshan County, Anhui Province, is located in the hinterland of Dabie Mountain, with the Dabie Mountain National Scenic Byway and the most beautiful tourism loop of Huoshan passing through the territory. The village is surrounded by mountains and beautiful scenery. The village has a total area of 1600 square kilometres, including 12,564 acres of mountains and forests, 1683 acres of arable land, and is rich in resources such as tea, Chinese herbs and silkworms. In the past, it was a typical mountainous village with poor economic development due to the high mountains and dense forests and closed traffic. In the past, it was a typical mountainous village with poor economic development due to the high mountains and dense forests and closed traffic. The Huoshan County government introduced Jiangsu Huaqiang Group, which invested 465 million USD to develop Steep Shahe Village in 2016, putting the village on the fast track to poverty alleviation through tourism, forming a high-end ecological tourism resort and hot spring town with distinctive self-characteristics, integrating tourism, leisure, holiday, entertainment, sports, health and retirement. At the same time, through the development of high quality organic agriculture and the use of financial means to promote consumption, the village has been able to achieve better poverty alleviation results in the short term, allowing this national rural tourism poverty alleviation focus village to regain momentum.

For the purpose of evolutionary game analysis, the key stakeholders were simplified to Huoshan County Government, Huaqiang Company and Steep Sha River residents, focusing on the spillover effects of Huoshan County Government, Huaqiang Company and Steep Sha River residents. Based on in-person investigations, and data collected from relevant government departments and enterprises, we used the following set of parameter values as the benchmark: the amount of equity is denoted by *W* = 10 million USD, the additional benefits will be recorded as *I* = 5 million USD, the total input of the enterprise is *F*_1_ = 40 million USD, when the local residents actively participate, the total return of the project is *E*_1_ = 50 million USD, the total return of the project is *E*_2_ = 45 million USD when the local residents passively participate, the total input of the enterprise is *F*_2_ = 20 million USD if the tourism enterprise adopts the strategy of “spurious rural ecotourism”, the total revenue of the project is *E*_3_ = 45 million USD when the local residents actively participate, the total revenue of the project is *E*_4_ = 35 million USD when the local residents negatively participate, the interest rate of *μ*_1_ = 0.2 if the government strictly regulates, the interest rate of *μ*_2_ = 0.1 if the government formally regulates, the basic wage for local residents to work in rural ecotourism enterprises is *S* = 5 million USD, *K*_1_ = 3 million USD is the bonus given to the active residents when the tourism enterprise takes a substantive action, and *K*_2_ = 1 million USD is the bonus given to the active residents when the tourism enterprise takes a spurious action, the income of *N* = 5 when the local residents negatively participating, *C*_1_ = 10 million USD indicating the publicity cost of the government with strict regulation and *C*_2_ = 5 million USD indicating the publicity cost of the government with formal regulation, *α*_1_ = 0.5 and *α*_2_ = 0.3 denote the publicity efforts of the government with strict regulation and formal regulation, *β* = 0.5 indicates the ability of local residents to understand the policy, the increase in benefits for local residents who actively participate under strict government regulation is indicated by *P* = 10 million USD, *H*_1_ = 10 million USD is the financial support of government with strict regulation to tourism enterprises, *H*_2_ = 5 million USD is the financial support of government with formal regulation to tourism enterprises, *C*_3_ = 5 million USD is the cost of monitoring of a government with strict regulation, *R* = 5 million USD is the praise reward given by the government to enterprises that substantially conduct rural ecotourism, *G =* 10 million USD is the fine imposed by the government on tourism enterprises that adopt spurious practices, *λ =* 0.5 is the probability of being detected by the government when tourism enterprises spuriously conduct rural ecotourism, *M* = 1 million USD indicates the reward that the government will give to the residents who actively participate, When the government “strictly regulates”, its credibility and reputation will be enhanced, which is recorded as *T* = 15 million USD. *x*_0_’ *y*_0_, *z*_0_ respectively, are the initial probabilities of substantial rural ecotourism by tourism enterprises, active participation by local residents and strict supervision by government.

(1)Strategy options for tourism enterprises in different initial states

Assuming the initial probability of local residents and government is 0.5, the strategy choices of tourism enterprises in different initial states are shown in the Figure 4, Figure 5 and Figure 6, where the red solid line indicates the strategy choice of tourism enterprises, the yellow dotted line indicates the strategy choice of local residents and the blue dashed line indicates the strategy choice of government. When the initial probability of a tourism enterprise choosing to substantially engage in rural ecotourism is 0.1, tourism enterprises are reluctant to take substantial action against local residents because of the high investment in rural ecotourism projects, the difficulty in generating short-term returns and the slow results. Thus local residents’ income from participating in rural ecotourism is not as high as their income from other activities, and they are reluctant to actively participate in rural tourism activities. In order to promote substantive rural eco-tourism of tourism enterprises and active participation of local residents, thereby achieving the goal of rural revitalization, the government will exercise strict regulation.

As the probability of a tourism enterprise initially choosing to engage in substantive rural ecotourism increases from 0.1 to 0.5 to 0.9, the willingness of tourism enterprises to engage in substantive rural ecotourism increases, leading to more income for local residents, who tend to change from passive participation to active participation. However, the rate at which tourism enterprises evolve into spuriously conducting rural ecotourism increases. This is due to the increase in the number of tourism enterprises joining the scheme and the increase in competition between them, which has led to a decrease in the revenue earned by tourism enterprises and therefore accelerated the trend towards sham rural ecotourism. At 0.5, the government still tends to choose strict regulation in order to promote local economic development. However, as the willingness of tourism enterprises to substantially engage in rural ecotourism increases, local residents are also willing to actively participate in it, which reduces the pressure of government regulation, so the government’s evolution to strict regulation slows down. At 0.9, the willingness of tourism enterprises to substantially engage in rural ecotourism is stronger, which drives local residents to actively participate in it, thus making government regulation less stressful and tending to opt for formal regulation.

(2)Strategy choices of local residents in different initial states

Assuming that the initial probability of tourism enterprises and the government is 0.5, the strategy choices of local residents in different initial states are shown in the Figure 7, Figure 8 and Figure 9, where the red solid line indicates the strategy choice of tourism enterprises, the yellow dotted line indicates the strategy choice of local residents, and the blue dashed line indicates the strategy choice of the government. When the probability of local residents initially choosing to actively participate is 0.1, some farmers have insufficient understanding and awareness of rural ecotourism, and many of them are accustomed to a long life of farming and are unwilling to change the status quo and stick to stereotypes, which leads to local residents tending to choose negative participation in rural ecotourism activities. The government will strictly regulate this in order to promote local residents out of poverty. Due to government policies and regulations, tourism companies are required to pay dividends and bonuses to local residents who participate in rural tourism activities, which increases the expenditure of tourism companies. At this time, the number of local residents willing to participate in rural tourism is relatively small, so tourism companies spend relatively little, and the government will support them financially and policy-wise. This will lead to a tendency for tourism enterprises to opt for substantial rural ecotourism.

As the probability of local residents initially choosing to actively participate increases from 0.1 to 0.5 to 0.9, tourism companies tend to choose to spuriously conduct rural ecotourism as the number of local residents actively participating increases and the total expenditure that tourism companies need to give to local residents increases because it’s not good for enterprises to make profits. The government still chooses to strictly regulate in order to improve performance and promote local economic development.

(3)The government’s choice of strategy in different initial states

Assuming an initial probability of 0.5 for tourism enterprises and local residents, the government’s strategy choices in different initial states are shown in the Figure 10, Figure 11 and Figure 12, where the red solid line indicates the strategy choice of tourism enterprises, the yellow dotted line indicates the strategy choice of local residents, and the blue dashed line indicates the government’s strategy choice. When the probability of the government initially choosing strict regulation is 0.1 and 0.5, tourism enterprises tend to choose the strategy of spuriously conducting rural ecotourism and local residents tend to choose the strategy of negative participation because the government does not provide sufficient financial support and encouragement to tourism enterprises and local residents, and tourism enterprises substantially conduct rural ecotourism and local residents actively participate without higher returns. In order to promote local economic development and rural revitalization, the government tends to opt for strict regulation.

When the probability of the government initially choosing strict regulation grows to 0.9, the degree of strict government regulation increases, tourism enterprises receive sufficient policy and financial support and are thus happy to substantially undertake rural ecotourism. It also results in more financial incentives for local residents who participate in it, and the government conducts better publicity and popularization of rural ecotourism policies, so that local residents understand the benefits of participating in rural tourism projects. As a result, local residents tend to choose to actively participate; the government tends to choose to strictly supervise in order to better realize rural revitalization.

(4)The impact of differences in government awareness and local residents’ ability to understand on the evolution of local residents’ strategies

In order to promote the smooth implementation of rural ecotourism activities, the government needs to publicize and popularize relevant policies, while residents in rural areas may not be aware of rural tourism projects due to their own understanding. 

Therefore, local residents do not choose to actively participate in a rural ecotourism project where government publicity is strong and local understanding is weak, as shown in the Figure 13, *α*_1_ = 0.9, *β* = 0.1, and *α*_1_ = 0.7, *β* = 0.3. When the government does not publicize rural ecotourism policies sufficiently, local residents have no access to relevant policy information, even if they have a high level of understanding, local residents tend to participate the rural ecotourism activities passively, as shown in figure *α*_1_ = 0.1, *β* = 0.9, and *α*_1_ = 0.3, *β* = 0.7. When the government’s efforts to publicize relevant policies and the ability of local residents to understand them are low, it is clear to see that local residents cannot be motivated to choose to actively participate, as shown in the figure *α*_1_ = 0.3, *β* = 0.3, and *α*_1_ = 0.5, *β* = 0.5. Only when the government increases the promotion and publicity of rural tourism-related policies, and local residents have a thorough understanding of the relevant policies, can the active participation of local residents be promoted, as shown in figure *α*_1_ = 0.7, *β* = 0.7. 

(5)The impact of differences in income from other activities of local residents on the evolution of local residents’ strategies

Local residents may engage in other activities while participating in rural ecotourism projects, such as selling agricultural and sideline products, making special handicrafts, etc. If the income from these activities is low, then it is less attractive to the residents, so local residents tend to choose to actively participate, as can be seen from the red solid line in the Figure 14. When the income from other activities is medium, although it will cause some attraction to local residents, it is still not enough to make them tend to choose to negatively participate in rural ecotourism activities, and the evolution of local residents to actively participate in rural ecotourism slows down, as shown by the yellow dotted line. When the income from other activities is high, then the local residents are more willing to engage in these activities rather than actively participate in rural tourism projects, as shown by the blue dashed line.

(6)The impact of the difference in the probability of being detected by the government when tourism enterprises engage in bogus rural ecotourism on the evolution of government strategies

Figure 15 shows the government’s strategy evolution under different probabilities of discovering that tourism enterprises engage in bogus rural ecotourism. Strict government regulation does not necessarily lead to the detection of fraudulent rural ecotourism of tourism enterprises. When the probability of detection is *λ* = 0.1, the probability of the government detecting a tourism company’s fraudulent conduct of rural ecotourism is low, but the cost of strict regulation is high, so the government is reluctant to impose strict regulation because it feels that it is costly and ineffective. At *λ =* 0.5, the probability of the government discovering that a tourism company is engaging in fraudulent rural ecotourism increases and the government tends to choose a strict regulation strategy. At *λ* = 0.9, the higher probability of a tourism company being found to falsely conduct rural ecotourism significantly increases the government’s incentive to strictly regulate, and the rate at which the government evolves to strictly regulate increases.

(7)The impact of differences in the probability of being detected by the government when tourism enterprises spuriously undertake rural ecotourism on the evolution of tourism enterprise strategies

Figure 16 shows the strategy evolution of tourism enterprises under different probabilities of being detected by the government. If the government does not find out and punish the tourism enterprises, the enterprises will be more willing to choose the strategy of spurious rural ecotourism. Because they can not only get the government’s financial support but also not pay more costs. At *λ* = 0.1, the probability of a tourism enterprise adopting a false practice being detected by the government is small, and tourism enterprises will tend to choose spurious for rural ecotourism at this time. At *λ* = 0.5, although the probability of a tourism enterprise falsely engaging in rural ecotourism being detected by the government increases, tourism enterprises still have a fluke mentality and tend to choose spurious acts, and their evolution into spurious acts slows down. At *λ* = 0.9, tourism companies have a higher probability of being detected for adopting spurious practices, at this time they tend to choose to substantially carry out rural ecotourism strategies.

From the above analysis: The different initial game states lead to different evolutionary stability strategy (ESS), with the final choice of which strategy depends on the probability of the various strategies initially chosen. In the long run, an ideal stable state is one in which tourism enterprises substantially undertake rural ecotourism, local residents actively participate, and the government strictly regulates. The government’s promotion of rural ecotourism policies is low, or local residents have a low understanding of the policies, local residents will choose to participate negatively. Local residents may engage in other activities while participating in rural ecotourism projects, such as selling agricultural and sideline products, making special handicrafts, etc. If the income from these activities is low, local residents will choose to actively participate in rural eco-tourism projects; if the income from these activities is high, local residents will choose to negatively participate in rural ecotourism projects. When the probability of a tourism enterprise engaging in false rural ecotourism is low and the government discovers this behavior, the tourism enterprise will choose the fake strategy and the government will choose the formal regulation; as the probability of discovery increases, the tourism enterprise will choose the substantial rural ecotourism strategy and the government will choose the strict regulation strategy. In this way, through the efforts of the Huoshan County Government, Huaqiang and the residents of Steep Shahe, Steep Shahe Village was transformed from a poor village into a high-end ecotourism resort. Since its opening in October 2017, Steep Shahe Hot Spring Town has received a total of more than one million visitors and achieved a comprehensive income of 77.56 million USD. Family hotels and farmhouse restaurants have blossomed in all aspects around Steep Shahe Village and the Shangtu City market town. In 2017 and 2018, 103 new farmhouses and farmhouse inns were added to Steep Shahe Village, and farm specialties such as tea leaves, local eggs, black-haired pigs, red lantern peppers and Steep Shahe vermicelli are very popular. The hot spring town has also achieved direct employment for farmers, with more than 500 migrant workers in the town working in the hot spring town, driving 86 households and 230 people in poverty to achieve stable employment, with an average household income of more than 4653.54 USD. The hot spring town and related projects have transferred a total of 1737 acres of land to 461 households, with annual transfer rents totalling 168,148.04 USD, directly increasing the income of farmers and ensuring that they are not unemployed and have guaranteed income. By combining the role of the organization with the role of the market, and the cooperative helping to sell with the farmers’ own sales, 50,000 kg of organic vermicelli have been sold by the cooperative since 2019, including 15,000 kg for 22 poor households, driving 180 households, including 45 poor households, with an average household income of 775.59 USD, and an increase of 31,023.62 USD in the village collective economy. During the epidemic, Steep Shahe Village helped 11 poor households sell 4515 kg of sweet potato vermicelli on their behalf, earning 21,010.75 USD, directly increasing the income of poor households and strongly contributing to poverty alleviation. Now, they are working towards the revitalization of the countryside in 2035.

## 5. Conclusions and Policy Recommendations

### 5.1. Conclusions

The following general conclusions are reached:(1)The different initial game states lead to different evolutionary stability strategy (ESS), with the final choice of which strategy depends on the probability of the various strategies initially chosen. In the long run, an ideal stable state is one in which tourism enterprises substantially undertake rural ecotourism, local residents actively participate, and the government strictly regulates.(2)When the government’s promotion of rural ecotourism policies is low, or local residents have a low understanding of the policies, local residents will choose to participate negatively. Only when the government’s support policies is high, and local residents have a thorough understanding of rural eco-tourism policies, can local residents be promoted to actively participate in rural eco-tourism activities and contribute to rural revitalization.(3)Local residents may engage in other activities while participating in rural ecotourism projects, such as selling agricultural and sideline products, making special handicrafts, etc. If the income from these activities is low, local residents will choose to actively participate in rural eco-tourism projects; if the income from these activities is high, local residents will choose to negatively participate in rural ecotourism projects.(4)When the probability of a tourism enterprise engaging in false rural ecotourism is low and the government discovers this behavior, the tourism enterprise will choose the fake strategy and the government will choose the formal regulation; as the probability of discovery increases, the tourism enterprise will choose the substantial rural ecotourism strategy and the government will choose the strict regulation strategy.

Besides, we can learn from this case. First, insist on the participation of the residents. Respect the residents, mobilize them, rely on them, and enhance the endogenous motivation of the rural masses to develop the countryside and achieve revitalization. The Government will also strengthen the internal motivation of rural people to develop the countryside and achieve revitalization. The establishment of an internal mechanism for the peasants to participate in the development of rural tourism on their own, giving them the right to have a full voice in economic activities, and promoting the development model of “small farmers, big industries” in rural economic activities with family management as the carrier, so that the peasants can start their own businesses and realize the prosperity of the people in the process of rural revitalization. Second, adhere to ecological tourism. Green water and green mountains are golden mountains. Taking the protection of the ecological environment of the countryside as the starting point, we will play the characteristic card, strengthen the deep integration of the characteristic town with tourism, ecology, service industry and agriculture, create a characteristic countryside that no one has, no one has, and no one is special, and improve the attractiveness of out-of-town businessmen and tourists. In the process of implementing the rural revitalization strategy and realizing the integrated development of urban and rural areas, the construction of special and strong, clustered and combined, refined and beautiful, living and new special rural tourism resorts. Through the development of eco-tourism, more and more people in Steep Shahe Village have been able to eat “tourism rice” and embark on the road to prosperity, which has greatly boosted the regional economic development and poverty alleviation, and laid a solid foundation for the implementation of the rural revitalization strategy. 

### 5.2. Theoretical Contribution

This research extends the current knowledge on the rural tourism industry with some important research dimensions. Most of the previous studies considered the rural tourism industry to be a simple view. The relevant studies on the problems and countermeasures in rural tourism [12,15,16], and study the rural tourism’ pathways and methods [54,55,56,57,58,59,60,61]. This study introduces the tripartite evolutionary game model and focuses on the tourism enterprises, local residents and government on rural ecotourism in terms of evolutionary game. Then, we use simulation method and case study to analyze the stability of interactions among the stakeholders and determine an equilibrium solution in the finite rationality case.

Some previous studies especially emphasized the evolutionary stable strategies of local governments, tourism enterprises and residents in the development and construction of ecotourism in ecologically fragile areas [32], and ignored the rural ecotourism among the three stakeholders in the poverty-stricken areas of China. In this research, we analyze how the different strategies of the three parties affect the rural ecotourism project and implications for rural revitalization. Based on the findings of the study, this paper provides countermeasures and recommendations for tourism enterprises, government, and local residents.

### 5.3. Practical Implications 

(1)For the government, it is necessary to increase the publicity and promotion of relevant rural ecotourism policies, so as to enhance local residents’ awareness of the relevant policies; increase the incentives for local residents of active participation; increase the incentives for enterprises that substantially implement rural ecotourism and the penalties for enterprises that falsely implement rural ecotourism; and improve the monitoring system to dynamically monitor the implementation of rural ecotourism projects by enterprises. It will increase the likelihood of fraudulent rural ecotourism projects being detected by tourism companies. The government should place rural revitalization in a prominent position, take the construction of a rural revitalization demonstration belt as an important grasp, plan carefully, highlight the characteristics according to local conditions, make full use of the linkage and radiation of the surrounding characteristic towns, and build a rural revitalization demonstration village with high standards. In addition, the local government should establish a fast and smooth information network, adopt modern information means such as e-commerce, negotiate projects through the Internet and other forms, and strive to expand online investment, while implementing entrusted investment or intermediary investment.(2)For tourism enterprises, the pursuit of profit maximize is their goal. However, rural ecotourism projects are heavily invested, and because returns are difficult and slow to achieve in the short term, tourism enterprises may be reluctant to undertake substantive implementation of rural ecotourism projects. If tourism enterprises falsely implement rural ecotourism projects, they will suffer a loss of overall social benefits. Therefore, tourism enterprises should also take social responsibility, have a long-term vision and respond to the government’s call to implement rural ecotourism projects for local residents, so as to better serve rural revitalization.(3)For local residents, they should change their traditional concepts, actively learn about the relevant rural ecotourism policies, and actively participate in rural ecotourism projects with the help of the government and tourism enterprises, so as to successfully realize their own poverty alleviation and re-employment, improve their own skills and meet the needs of rural revitalization. At the same time, based on the actual demand for the development and operation of rural eco-tourism resources, local residents are provided with skills training in the dissemination of rural eco-tourism cultural resources, the transmission of special skills and the operation of B&Bs, so as to expand the audience for vocational education and strengthen the effectiveness of vocational education.

### 5.4. Limitations and Future Research

Although this study has significant theoretical and practical implications, it also has several limitations that could be explored in future research. The first is that the parameters of the evolutionary game payment matrix do not take into account all the factors affecting rural ecotourism. And the second is that some of the parameters of the payment matrix may not be precisely assigned due to the constraints of the survey respondents and conditions, which is where further research should be conducted in the future. In addition, other players are important subjects that should be considered when establishing the evolutionary game model. Future research may investigate more stakeholders or consider the technology innovation, innovation ecosystem with the poverty areas to ecotourism

## Figures and Tables

**Figure 1 ijerph-18-09690-f001:**
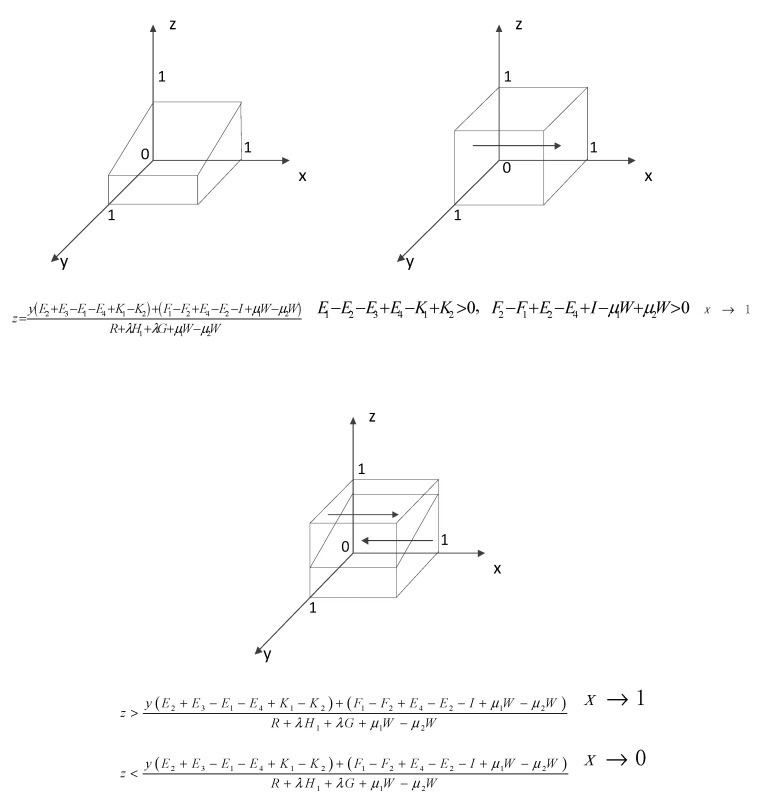
Dynamic evolution of tourism enterprises’ behavioral choice strategies.

**Figure 2 ijerph-18-09690-f002:**
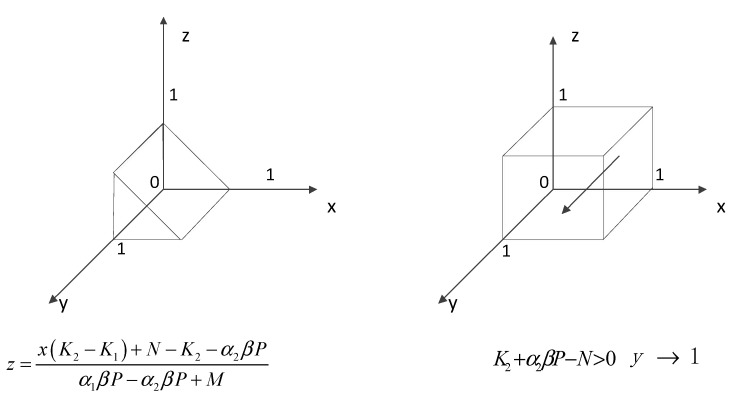
Dynamic evolution of local residents’ behavioral choice strategies.

**Figure 3 ijerph-18-09690-f003:**
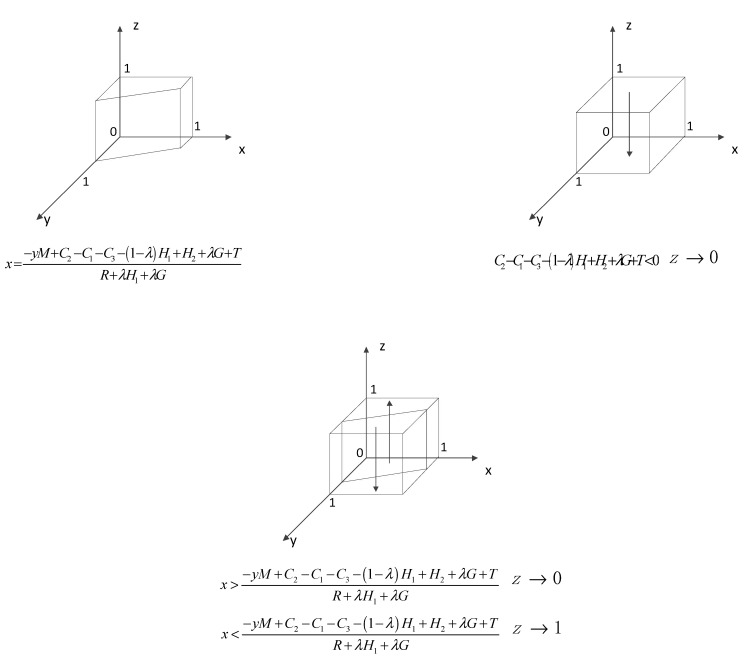
Dynamic evolution of the government’s behavioral choice strategies.

**Figure 4 ijerph-18-09690-f004:**
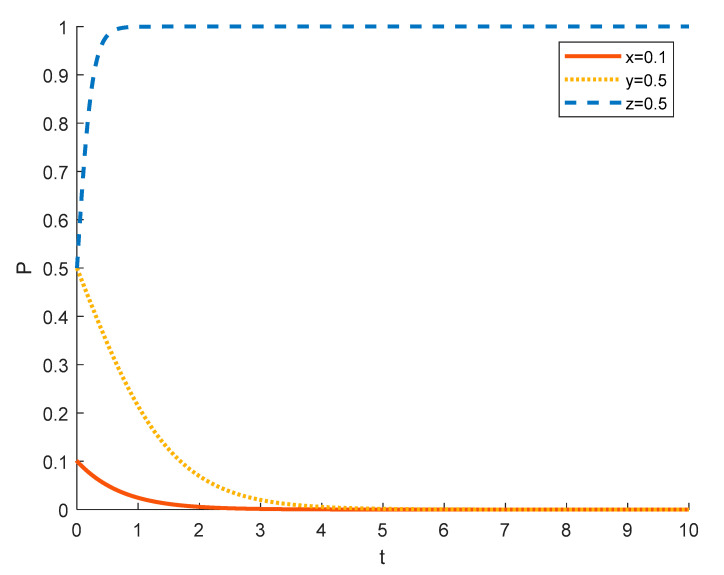
Strategy evolution when the initial value of *x* is 0.1.

**Figure 5 ijerph-18-09690-f005:**
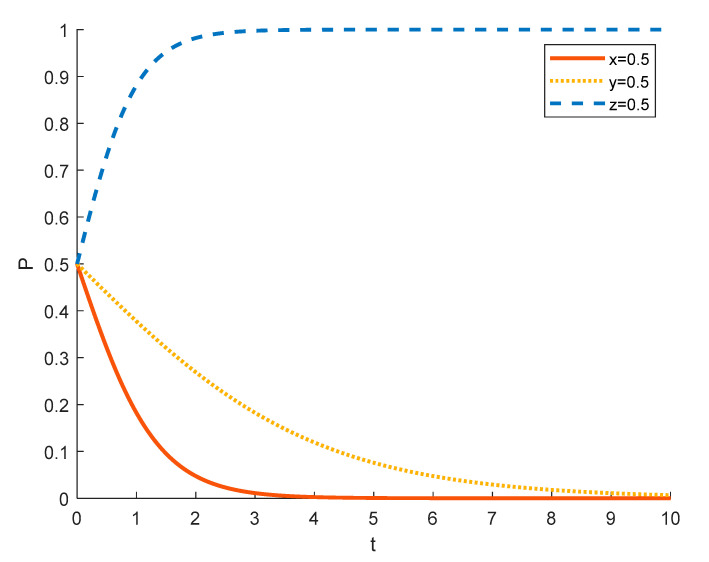
Strategy evolution when the initial value of *x* is 0.5.

**Figure 6 ijerph-18-09690-f006:**
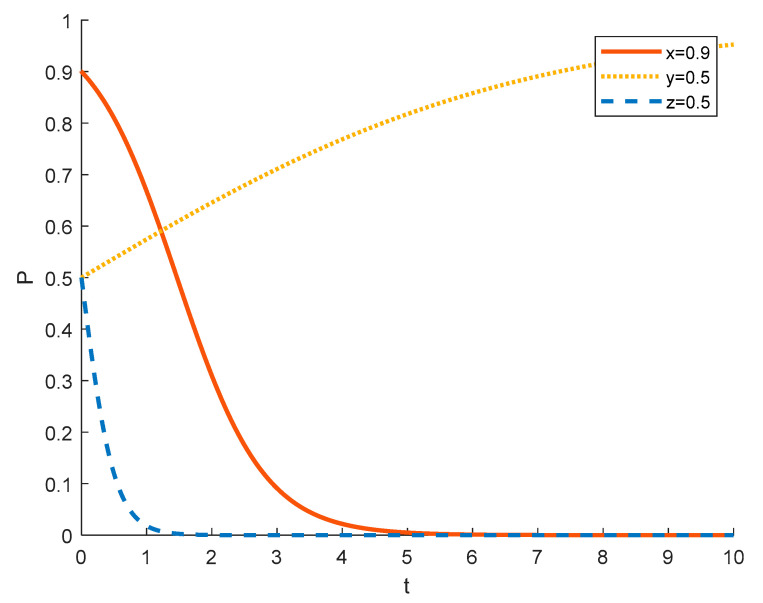
Strategy evolution when the initial value of *x* is 0.9.

**Figure 7 ijerph-18-09690-f007:**
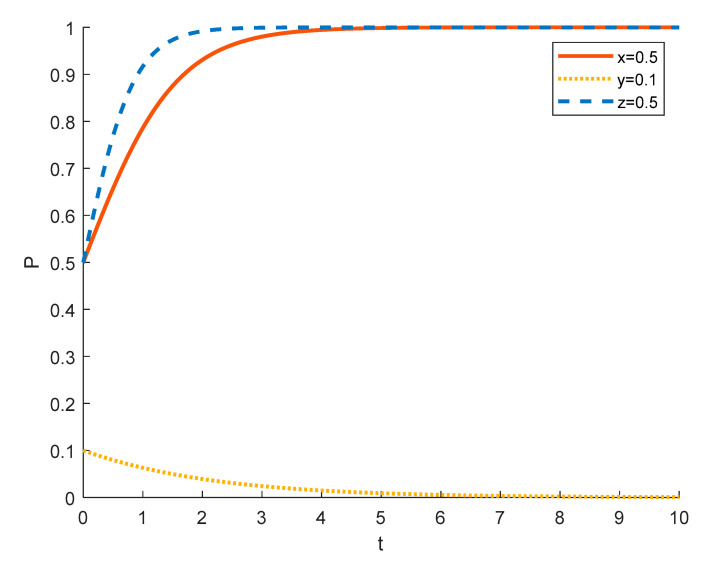
Strategy evolution when the initial value of *y* is 0.1.

**Figure 8 ijerph-18-09690-f008:**
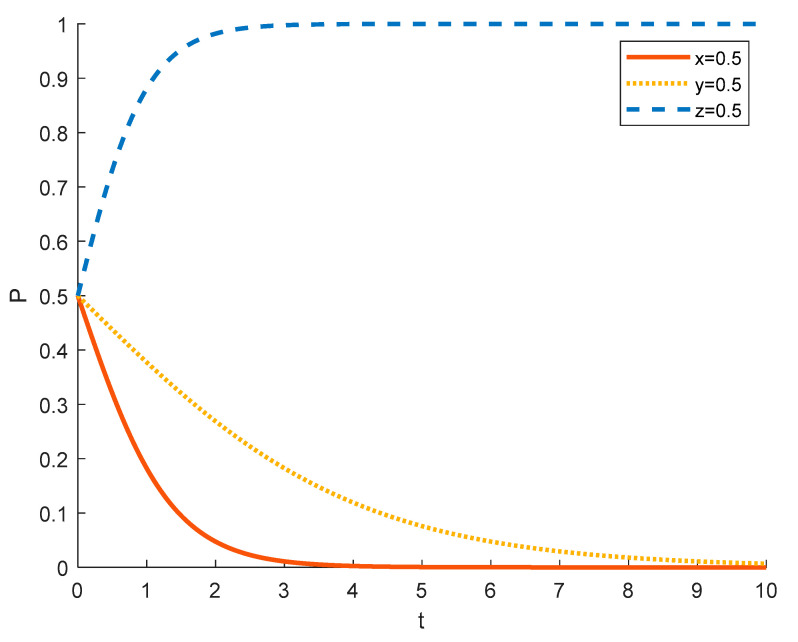
Strategy evolution when the initial value of *y* is 0.5.

**Figure 9 ijerph-18-09690-f009:**
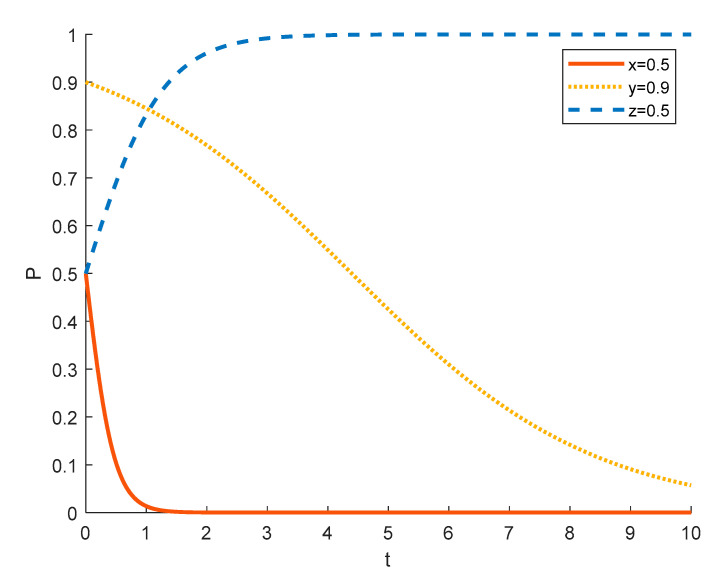
Strategy evolution when the initial value of *y* is 0.9.

**Figure 10 ijerph-18-09690-f010:**
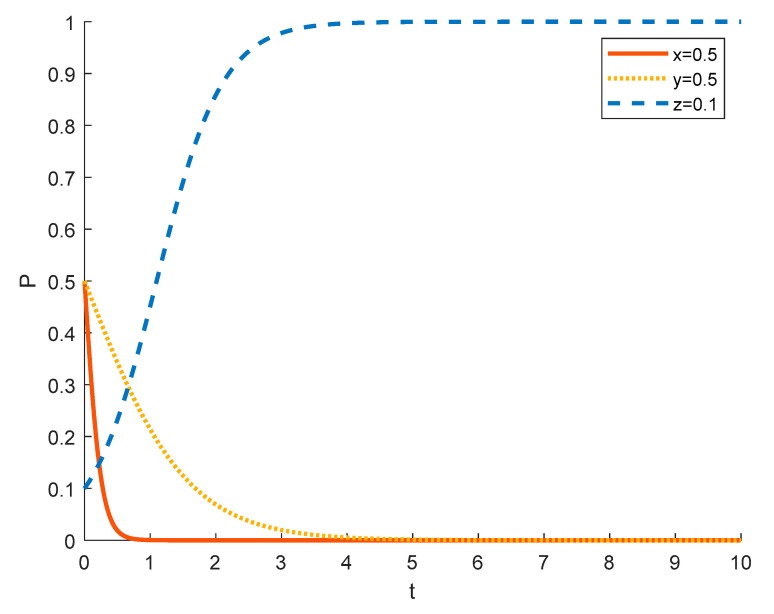
Strategy evolution when the initial value of *z* is 0.1.

**Figure 11 ijerph-18-09690-f011:**
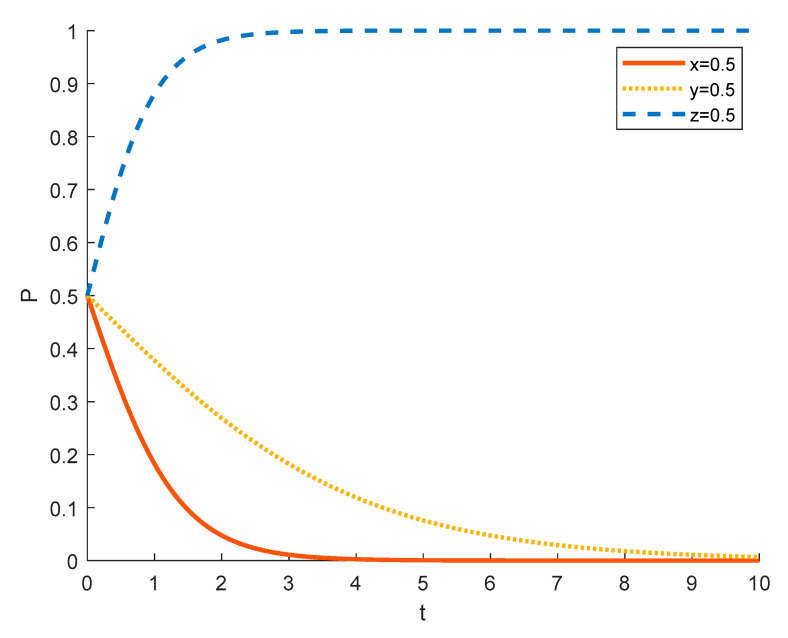
Strategy evolution when the initial value of *z* is 0.5.

**Figure 12 ijerph-18-09690-f012:**
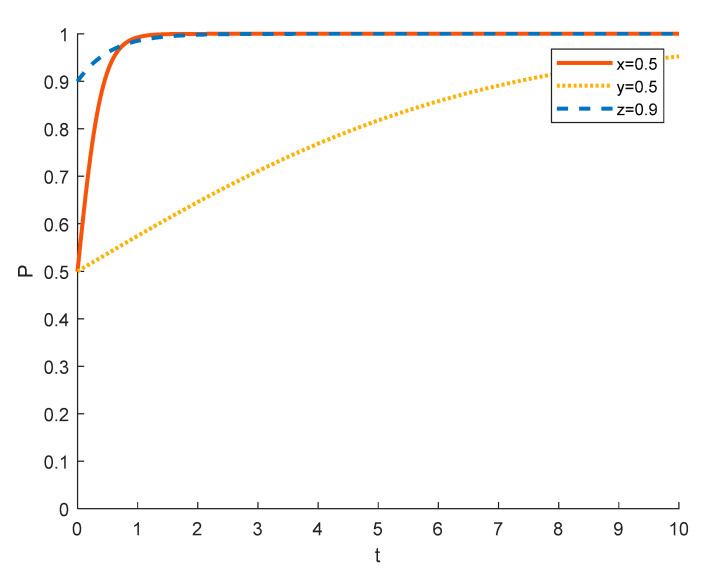
Strategy evolution when the initial value of *z* is 0.9.

**Figure 13 ijerph-18-09690-f013:**
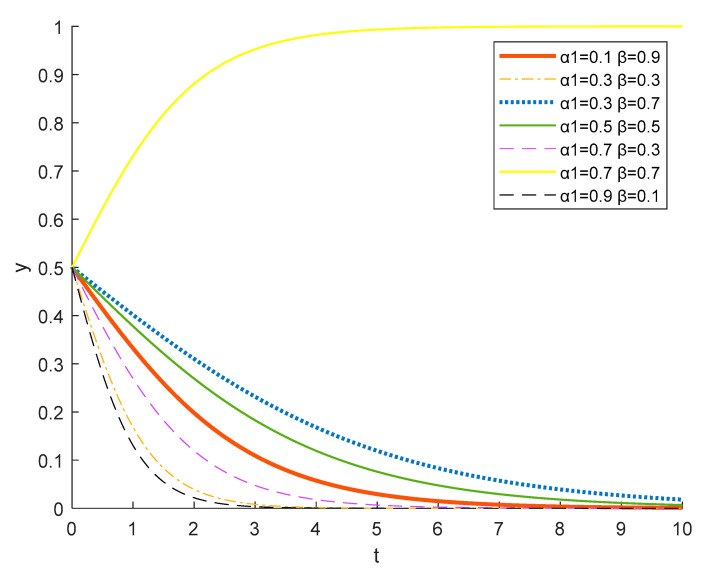
Strategy evolution of local residents under different publicity efforts of the government and different understanding abilities of local residents.

**Figure 14 ijerph-18-09690-f014:**
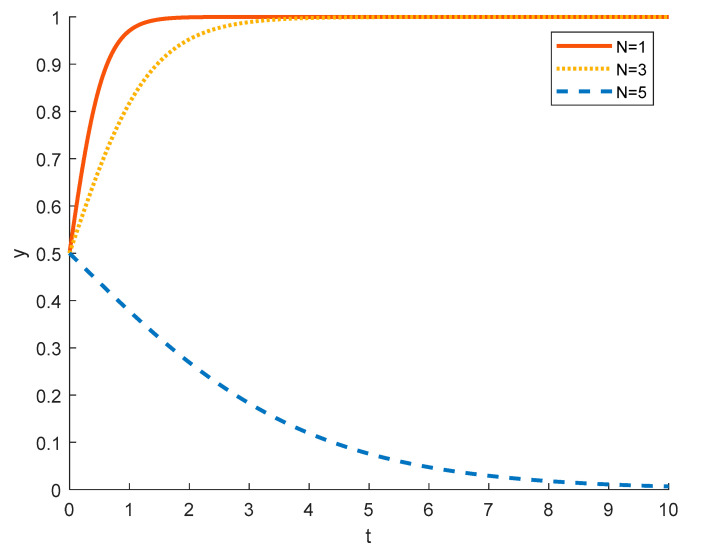
Strategy evolution of local residents with different income from other activities.

**Figure 15 ijerph-18-09690-f015:**
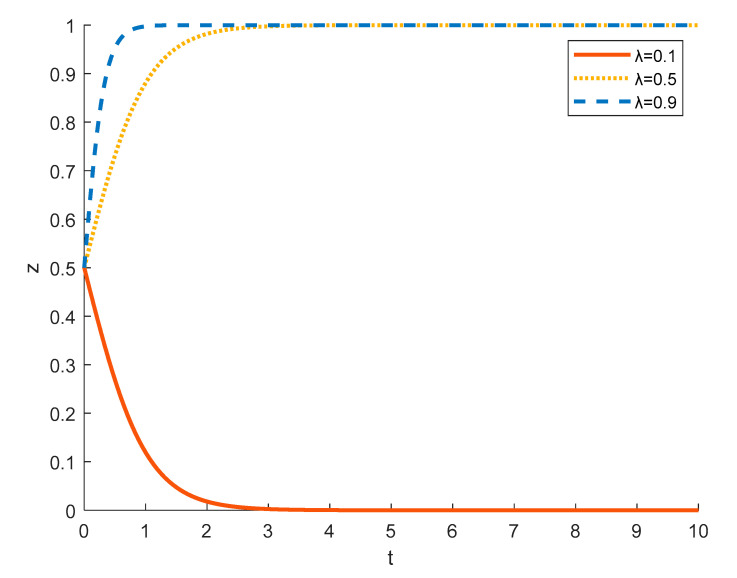
Government’s strategy evolution under different discovery probabilities.

**Figure 16 ijerph-18-09690-f016:**
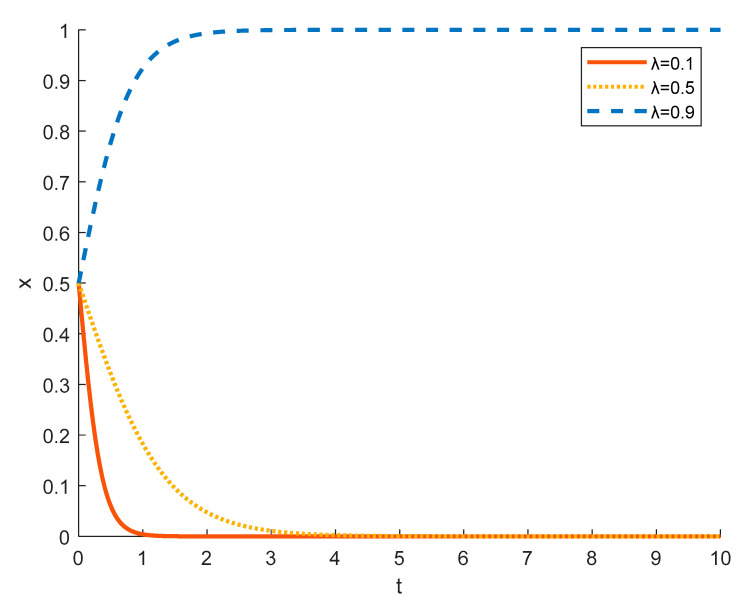
Strategy evolution of tourism enterprises under different discovery probabilities.

**Table 1 ijerph-18-09690-t001:** The payoff matrix of evolutionary game.

		Local Residents	
Tourism enterprise substantive	Government strict regulation	Active participation $E_{1}+I-F_{1}-\mu_{1}W-S-K_{1}+\Delta F+H_{1}+R$ $\mu_{1}W+S+K_{1}+\alpha_{1}\beta P+M$ $-C_{1}-H_{1}-C_{3}-R-M+T+L$	Negative participation $E_{2}+I-F_{1}-\mu_{1}W-S+\Delta F+H_{1}+R$ $\mu_{1}W+S+N$ $-C_{1}-H_{1}-C_{3}-R+T+L$
	Government formal regulation	$E_{1}+I-F_{1}-\mu_{1}W-S-K_{1}+\Delta F+H_{2}$ $\mu_{1}W+S+K_{1}+\alpha_{2}\beta P$ $-C_{2}-H_{2}+L$	$E_{2}+I-F_{1}-\mu_{1}W-S+\Delta F+H_{2}$ $\mu_{1}W+S+N$ $-C_{2}-H_{2}+L$
Tourism enterprise spurious	Government strict regulation	$E_{3}-F_{2}-\mu_{1}W-S-K_{2}+\Delta F+\left(1-\lambda \right)H_{1}-\lambda G$ $\mu_{1}W+S+K_{2}+\alpha_{1}\beta P+M$ $-C_{1}-C_{3}-\left(1-\lambda \right)H_{1}+\lambda G-M+T-D$	$E_{4}-F_{2}-\mu_{1}W-S+\Delta F+\left(1-\lambda \right)H_{1}-\lambda G$ $\mu_{1}W+S+N$ $-C_{1}-C_{3}-\left(1-\lambda \right)H_{1}+\lambda G+T-D$
	Government formal regulation	$E_{3}-F_{2}-\mu_{2}W-S-K_{2}+\Delta F+H_{2}$ $\mu_{2}W+S+K_{2}+\alpha_{2}\beta P$ $-C_{2}-H_{2}-D$	$E_{4}-F_{2}-\mu_{2}W-S+\Delta F+H_{2}$ $\mu_{2}W+S+N$ $-C_{2}-H_{2}-D$

## Data Availability

Data is contained within the article.

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
