# Peer review of "The Regime of Rural Ecotourism Stakeholders in Poverty-Stricken Areas of China: Implications for Rural Revitalization"

_ijerph, 2021, doi:10.3390/ijerph18189690_

Round 1
Reviewer 1 Report
Title: The regime of rural ecotourism stakeholders in poverty-stricken areas of China: implications for rural revitalization
The goal of the article is to analyses the relationship among the tourism enterprises, local residents and the government on rural ecotourism. Authors proposed the evolution game model of rural ecotourism and analyzed the results.
General comment:
The positive of this manuscript is that it has related the model to the cases and proposed the poverty areas as the promotion targets. However, the authors do not clearly distinguish the differences of the terms such as ecotourism, rural tourism, rural ecotourism and tourism. Lots of terms in the manuscript are mixing up, I suggest in the section of introduction, the relative terms should be reviewed. Besides, this manuscript does not propose a detailed practice which can apply to, all the conclusions are general and lack of operability. I suggest authors review the references which concerning on stakeholders of tourism recently, especially the model analysis. Meanwhile, the identification of the stakeholders should be clearly explained because they are the key concerns of the model. Lastly, even I am not a native speaker, I recommend find some native speaker to polish the English writing.
Particular comments:
In the section of Introduction, there are lots of phrases in quotation, Line 35, Line 45, Line 47-48, Line 51, Line 56. Line 64 etc. making the section very confuse. The readers are not understanding the meaning of these phrases.
The section 1. Introduction and 2. Literature review should be grouped together as one section.
Line 90: the term “false” may not suitable here and in following sections, please review the whole manuscript.
Line 139 and Line 149: the abbreviations should be as full term when first described. Please review the whole manuscript.
Section 3.2: I don’t think this section proposed any scientific hypotheses, they are the description of the model construction. The hypotheses should be proposed in the Introduction and tested follow on. Actually, this manuscript is lacking a reasonable scientific hypothesis.
The section number has mistaken, “3. Model Building and Analysis” and “3. Case Analysis and Numerical Simulation” used the same number.
Section 3.3: there are too much space describing the model, I suggest put the detailed parts in the appendix but not the main text. And weather the inference of the model needs the support from real data, in this version, I didn’t find any real data in model construction.
From Line 481, Section 4 (should be) Case Analysis and Numerical Simulation: the outcomes of the model are the results, in the current version, hardly sum up the key points. Too much figures confused the readers. I suggest reconstructing this section, simplifying or grouping the figures, if necessary, use the statistical methods to describe your results.
Line 490: the unit for areas is correct? Why you used different units?
Line 496: “3 billion yuan” means Chinese Yuan (CNY) I suppose? I suggest transfer the currency to US dollars since you are publishing the article in international journal.
Line 749-753: no need to repeat the process you have mentioned in the previous sections.
Section 5.1: the four points are general conclusions, need to related the data you got from the model to do the further analysis and discussion.
Section 5.2: the tripartite relationship is not a novel concern in recent studies, there are many references discuss about it. Relating the poverty areas to ecotourism is an interesting idea, however, should consider the potential for implement.
Section 5.3: As previous comments, the ideas are general, the practical suggestions should come from the real data and analysis with detailed description.
Author Response
To Reviewer
Dear Reviewer,
Thank you for your letter and for the reviewer’ comments concerning our manuscript entitled “The regime of rural ecotourism stakeholders in poverty-stricken areas of China: implications for rural revitalization”(ID: ijerph-1323897).
Those comments are all valuable and helpful for revising and improving our paper, as well as the important guiding significance to our researches. We have studied comments carefully and have made correction which we hope meet with approval. Revised portions are marked by track changes in the “revised manuscript”. The main corrections in the paper and the responds to the reviewer’s comments are as follows.
Responds to the reviewer’s comments 1:
- The positive of this manuscript is that it has related the model to the cases and proposed the poverty areas as the promotion targets. However, the authors do not clearly distinguish the differences of the terms such as ecotourism, rural tourism, rural ecotourism and tourism. Lots of terms in the manuscript are mixing up, I suggest in the section of introduction, the relative terms should be reviewed.
Answer:
Thank you very much for pointing out the problem. We have revised our introduction and gave a definition of the terms such as ecotourism, rural tourism and rural ecotourism. Revised portions are marked in red in the “revised manuscript”. As following:
Rural Tourism has long been recognized in certain parts of Europe as an effective catalyst of rural socio-economic regeneration for over a hundred years [45]. Along with the development of rural tourism worldwide, rural tourism concept has many interpretations. Tourism activity in rural areas has remarkably increased in all the developed countries and developing countries worldwide, which has played a key role in the development of rural areas that were economically and socially depressed [46-47].
[45] Jing-Ming, H. E. (2003). A Study on Rural Tourism Overseas. Tourism Tribune, 18(1), 76-80.
[46] Yagüe, P. (2002). Rural tourism in Spain. Annals of tourism Research, 29(4), 1101-1110.
[47] Fons, M. V. S., Fierro, J. A. M., & y Patiño, M. G. (2011). Rural tourism: A sustainable alternative. Applied Energy, 88(2), 551-557.
The term ecotourism is surrounded by confusion [64]. It has been defined as ‘‘responsible travel to natural areas that conserves the environment and sustains the well-being of local people [65]’’. However, it is contended here that, regardless of definition, ecotourism is an instigator of change. It is inevitable that the introduction of tourists to areas that were previously seldom visited by outsiders will place new demands upon the environment associated with new actors, new activities, and new facilities. This will involve the forging of new relationships between people and environment, between peoples with different life-styles, and between a wide variety of forces for both change and stability. These forces act at a diversity of scales from global to local. Change is desired by most of the players involved in ecotourism, many of whom would like to see what they regard as an improvement in the existing situation. Rural ecotourism is a kind of tourism that takes the protection of the natural ecological environment as the premise and relies on the good natural ecological environment and unique human resources in the countryside to carry out ecological experience, ecological education, ecological cognition, while obtaining physical and mental enjoyment [66].
[64] Cater, E. (1994). Ecotourism in the Third World: problems and prospects for sustainability. Ecotourism: a sustainable option?., 69-86.
[65] Blangy, S., & Wood, M. E. (1993). Developing and implementing ecotourism guidelines for wildlands and neighboring communities (pp. 32-54). Ecotourism Society.
[66] Xiang, C., & Yin, L. (2020). Study on the rural ecotourism resource evaluation system. Environmental Technology & Innovation, 20, 101131.
- Besides, this manuscript does not propose a detailed practice which can apply to, all the conclusions are general and lack of operability. I suggest authors review the references which concerning on stakeholders of tourism recently, especially the model analysis. Meanwhile, the identification of the stakeholders should be clearly explained because they are the key concerns of the model. Lastly, even I am not a native speaker, I recommend find some native speaker to polish the English writing.
Answer:
Thank you very much for pointing out the problem. We have added the references. Also, we have explained the stakeholders. At the same time, we have polished the whole manuscript. Revised portions are marked in red in the “revised manuscript”. As following:
[12] Liu, C., Dou, X., Li, J., & Cai, L. A. (2020). Analyzing government role in rural tourism development: An empirical investigation from China. Journal of Rural Studies, 79, 177-188.
[32] Wang, W., Feng, L., Zheng, T., & Liu, Y. (2021). The sustainability of ecotourism stakeholders in ecologically fragile areas: implications for cleaner production. Journal of Cleaner Production, 279, 123606.
[36]Su, M. M., Wall, G., & Ma, Z. (2014). Assessing ecotourism from a multi-stakeholder perspective: Xingkai lake national nature reserve, China. Environmental management, 54(5), 1190-1207.
[41] Diamantis, D. (2018). Stakeholder ecotourism management: exchanges, coordination’s and adaptations. JOURNAL OF ECOTOURISM, 17(3), 203-205.
[42] Wondirad, A., Tolkach, D., & King, B. (2020). Stakeholder collaboration as a major factor for sustainable ecotourism development in developing countries. Tourism Management, 78, 104024.
[66] Xiang, C., & Yin, L. (2020). Study on the rural ecotourism resource evaluation system. Environmental Technology & Innovation, 20, 101131.
[82] Bu, Y., Wang, E., & Yu, Y. (2021). Analysis on asymptotic stabilization of eco-compensation program for forest ecotourism stakeholders. Environmental Science and Pollution Research, 1-17.
3.1. Analysis of the interest issues of each stakeholder
There are three main game stakeholders in rural ecotourism: tourism enterprises involved in rural ecotourism, local residents and the government. The strategies that tourism enterprises can choose are substantive rural ecotourism and spurious rural ecotourism, the strategies that local residents can choose are active participation and negative participation, and the strategies that the government can choose are strict regulation and formal regulation. There are the following game relations among stakeholders.
- The game between government and tourism enterprises
The government is the supporter, participant and supervisor of rural ecotourism. There are two strategies of government in the evolutionary game of rural ecotourism: strict regulation and formal regulation. Under strict regulation, the government will support tourism enterprises with policies and provide sufficient financial support, and reward and punish tourism enterprises for substantial and false spurious behavior, respectively. And the government will recover financial support for enterprises that spuriously conduct rural ecotourism. Conversely, if the government carries out formal regulation, it will be inactive in the process of rural ecotourism in most cases, and provide policy support but only less funding to tourism enterprises. Depending on the government's different strategies, tourism enterprises will constantly adjust and thus make the appropriate strategic choices.
- The game between government and local residents
The government with strict supervision vigorously publicizes policies related to rural ecotourism, thus encouraging local residents to actively participate in rural tourism projects and rewarding those who do so. The government with formal supervision, on the other hand, publicizes the relevant policies less vigorously and does not offer any rewards to local residents for their active participation. Local residents are guided by the government and choose their own strategies based on their understanding of the policies.
- The Game between tourism enterprises and local residents
Local residents will take shares in rural tourism projects with assets such as land and houses, and work for tourism enterprises. The tourism enterprises that substantively conduct rural ecotourism will pay dividends to local residents at a certain interest rate, take on some local residents to work in their tourism projects, and pay them basic wages and bonuses. Conversely, tourism enterprises that spuriously conduct rural ecotourism rarely take on local residents, and if they do, they only give them a small bonus. Local residents may earn income from other business activities while participating in rural ecotourism projects. As a result, local residents make adjustments and strategic choices based on the income they receive and the company's attitude toward them.
- In the section of Introduction, there are lots of phrases in quotation, Line 35, Line 45, Line 47-48, Line 51, Line 56. Line 64 etc. making the section very confuse. The readers are not understanding the meaning of these phrases. The section 1. Introduction and 2. Literature review should be grouped together as one section.
Answer:
Thank you very much for pointing out the problem. We have revised our introduction and literature review. Revised portions are marked in red in the “revised manuscript”. As following:
such as, Manaus in Brazil, Cancun Island in Mexico, Las Vegas in the United States, the Baleares Islands in Spain, the French Alps, Phuket in Thailand, Bali in Indonesia, the Sinai Peninsula in Egypt and Morocco in Africa, which were previously very backward places, have all become world-famous tourist destinations as a result of rural tourism development, with great social, economic and cultural improvement and development. In particular, as highlighted by other authors [10-11], the reinforcement of local food productions and short food supply chains might play a key role in rural ecotourism revitalization.
Therefore, it is necessary to study the stakeholders’ in interaction and strategic choice in the development and construction of rural ecotourism in poverty-stricken areas [41-43]. In our study, we plan an ecotourism system and analysis the relationship of the tourism enterprises, local residents and government by using an evolutionary game theory. Based on the theoretical analysis, an evolution game model for the three stakeholders is developed and the evolution process of strategies is described by replicator dynamic equations [32, 44]. Then, we use simulation method and case study to analyze the stability of interactions among the stakeholders and determine an equilibrium solution in the finite rationality case. Finally, specific control strategies are proposed to suppress instability and an ideal evolutionarily stable strategy is obtained. This provides a theoretical basis for achieving a win-win situation among the three parties. Our paper has two contributions that are not fully addressed in the previous literature. Firstly, we focus on the tourism enterprises, local residents and government on rural ecotourism in terms of evolutionary game, and analyze the regime mechanism of rural ecotourism among the three stakeholders in the poverty-stricken areas of China. Secondly, we use case study and simulation to analyze how the different strategies of the three parties affect the rural ecotourism project and implication for rural revitalization. Based on the findings of the study, this paper provides countermeasures and recommendations for tourism enterprises, government, and local residents.
Rural Tourism has long been recognized in certain parts of Europe as an effective catalyst of rural socio-economic regeneration for over a hundred years [45]. Along with the development of rural tourism worldwide, rural tourism concept has many interpretations. Tourism activity in rural areas has remarkably increased in all the developed countries and developing countries worldwide, which has played a key role in the development of rural areas that were economically and socially depressed [46-47].
The term ecotourism is surrounded by confusion [64]. It has been defined as ‘‘re-sponsible travel to natural areas that conserves the environment and sustains the well-being of local people [65]’’. However, it is contended here that, regardless of definition, ecotourism is an instigator of change. It is inevitable that the introduction of tourists to areas that were previously seldom visited by outsiders will place new demands upon the environment associated with new actors, new activities, and new facilities. This will involve the forging of new relationships between people and environment, between peoples with different life-styles, and between a wide variety of forces for both change and stability. These forces act at a diversity of scales from global to local. Change is desired by most of the players involved in ecotourism, many of whom would like to see what they regard as an improvement in the existing situation. Rural ecotourism is a kind of tourism that takes the protection of the natural ecological environment as the premise and relies on the good natural ecological environment and unique human resources in the countryside to carry out ecological experience, ecological education, ecological cognition, while obtaining physical and mental enjoyment [66].
- Line 90: the term “false” may not suitable here and in following sections, please review the whole manuscript.
Answer:
Thank you very much for pointing out the problem. We have changed the term “false” to the term “spurious” in our paper. Revised portions are marked in red in the “revised manuscript”.
- Line 139 and Line 149: the abbreviations should be as full term when first described. Please review the whole manuscript.
Answer:
Thank you very much for pointing out the problem. We have changed the abbreviations when first described in line 139 and line 149. Revised portions are marked in red in the “revised manuscript”. As following:
O-RHB (Organization-Resource-Humanity-Benefit). DEA (Data Envelopment Analysis).
- Section 3.2: I don’t think this section proposed any scientific hypotheses, they are the description of the model construction. The hypotheses should be proposed in the Introduction and tested follow on. Actually, this manuscript is lacking a reasonable scientific hypothesis.
Answer:
Thank you very much for pointing out the problem. We have revised the section 3.2. Revised portions are marked in red in the “revised manuscript”.
There is a difference between the basic game assumptions and the assumptions of empirical studies. In a finite rational environment, we give the stakeholders some parameters and then analyze their evolutionary state.
- The section number has mistaken, “3. Model Building and Analysis” and “3. Case Analysis and Numerical Simulation” used the same number.
Answer:
Thank you very much for pointing out the problem. We have revised our mistake. Revised portions are marked in red in the “revised manuscript”. As following:
- Case Analysis and Numerical Simulation
- Section 3.3: there are too much space describing the model, I suggest put the detailed parts in the appendix but not the main text. And weather the inference of the model needs the support from real data, in this version, I didn’t find any real data in model construction.
Answer:
Thank you very much for pointing out the problem. We have revised this section. Revised portions are marked in red in the “revised manuscript”. As following:
The section 3.3 is the normal reasoning logic of the evolutionary game. We have tried to streamline the process as much as possible. The data of our research from the section of case analysis and numerical simulation.
- From Line 481, Section 4 (should be) Case Analysis and Numerical Simulation: the outcomes of the model are the results, in the current version, hardly sum up the key points. Too much figures confused the readers. I suggest reconstructing this section, simplifying or grouping the figures, if necessary, use the statistical methods to describe your results.
Answer:
Thank you very much for pointing out the problem. We have revised this section. Revised portions are marked in red in the “revised manuscript”. As following:
From the above analysis: The different initial game states lead to different evolutionary stability strategy (ESS), with the final choice of which strategy depends on the probability of the various strategies initially chosen. In the long run, an ideal stable state is one in which tourism enterprises substantially undertake rural ecotourism, local residents actively participate, and the government strictly regulates. The government’s promotion of rural ecotourism policies is low, or local residents have a low understanding of the policies, local residents will choose to participate negatively. Local residents may engage in other activities while participating in rural ecotourism projects, such as selling agricultural and sideline products, making special handicrafts, etc. If the income from these activities is low, local residents will choose to actively participate in rural eco-tourism projects; if the income from these activities is high, local residents will choose to negatively participate in rural ecotourism projects. When the probability of a tourism enterprise engaging in false rural ecotourism is low and the government discovers this behavior, the tourism enterprise will choose the fake strategy and the government will choose the formal regulation; as the probability of discovery increases, the tourism enterprise will choose the substantial rural ecotourism strategy and the government will choose the strict regulation strategy.
- Line 490: the unit for areas is correct? Why you used different units?
Answer:
Thank you very much for pointing out the problem. We have revised this section. Revised portions are marked in red in the “revised manuscript”. As following:
The village is surrounded by mountains and beautiful scenery. The village has a total area of 1,600 square kilometres, including 12,564 acres of mountains and forests, 1,683 acres of arable land, and is rich in resources such as tea, Chinese herbs and silkworms.
- Line 496: “3 billion yuan” means Chinese Yuan (CNY) I suppose? I suggest transfer the currency to US dollars since you are publishing the article in international journal.
Answer:
Thank you very much for pointing out the problem. We have revised this section. Revised portions are marked in red in the “revised manuscript”. As following:
The Huoshan County government introduced Jiangsu Huaqiang Group, which invested 465 million USD to develop Steep Shahe Village in 2016,
For the purpose of evolutionary game analysis, the key stakeholders were simplified to Huoshan County Government, Huaqiang Company and Steep Sha River residents, focusing on the spillover effects of Huoshan County Government, Huaqiang Company and Steep Sha River residents. Based on in-person investigations, and data collected from relevant government departments and enterprises, we used the following set of parameter values as the benchmark: the amount of equity is denoted bymillion USD, the additional benefits will be recorded as million USD, the total input of the enterprise is million USD, when the local residents actively participate, the total return of the project is million USD, the total return of the project ismillion USD when the local residents passively participate, the total input of the enterprise is million USD if the tourism enterprise adopts the strategy of “spurious rural ecotourism”, the total revenue of the project is million USD when the local residents actively participate, the total revenue of the project is million USD when the local residents negatively participate, the interest rate of if the government strictly regulates, the interest rate of if the government formally regulates, the basic wage for local residents to work in rural ecotourism enterprises is million USD, million USD is the bonus given to the active residents when the tourism enterprise takes a substantive action, and million USD is the bonus given to the active residents when the tourism enterprise takes a spurious action, the income of when the local residents negatively participating, million USD indicating the publicity cost of the government with strict regulation and million USD indicating the publicity cost of the government with formal regulation, and denote the publicity efforts of the government with strict regulation and formal regulation, indicates the ability of local residents to understand the policy, the increase in benefits for local residents who actively participate under strict government regulation is indicated by million USD, million USD is the financial support of government with strict regulation to tourism enterprises, million USD is the financial support of government with formal regulation to tourism enterprises, million USD is the cost of monitoring of a government with strict regulation,million USD is the praise reward given by the government to enterprises that substantially conduct rural ecotourism, million USD is the fine imposed by the government on tourism enterprises that adopt spurious practices, is the probability of being detected by the government when tourism enterprises spuriously conduct rural ecotourism, million USD indicates the reward that the government will give to the residents who actively participate, When the government “strictly regulates”, its credibility and reputation will be enhanced, which is recorded as million USD. ’ respectively, is the initial probabilities of substantial rural ecotourism by tourism enterprises, active participation by local residents and strict supervision by government.
Since its opening in October 2017, Steep Shahe Hot Spring Town has received a total of more than one million visitors and achieved a comprehensive income of 77.56 million USD. Family hotels and farmhouse restaurants have blossomed in all aspects around Steep Shahe Village and the Shangtu City market town. In 2017 and 2018, 103 new farmhouses and farmhouse inns were added to Steep Shahe Village, and farm specialties such as tea leaves, local eggs, black-haired pigs, red lantern peppers and Steep Shahe vermicelli are very popular. The hot spring town has also achieved direct employment for farmers, with more than 500 migrant workers in the town working in the hot spring town, driving 86 households and 230 people in poverty to achieve stable employment, with an average household income of more than 4653.54 USD. The hot spring town and related projects have transferred a total of 1,737 acres of land to 461 households, with annual transfer rents totalling 168148.04 USD, directly increasing the income of farmers and ensuring that they are not unemployed and have guaranteed income. By combining the role of the organization with the role of the market, and the cooperative helping to sell with the farmers' own sales, 50,000 kg of organic vermicelli have been sold by the cooperative since 2019, including 15,000 kg for 22 poor households, driving 180 households, including 45 poor households, with an average household income of 775.59 USD, and an increase of 31023.62 USD in the village collective economy. During the epidemic, Steep Shahe Village helped 11 poor households sell 4515 kg of sweet potato vermicelli on their behalf, earning 21010.75 USD, directly increasing the income of poor households and strongly contributing to poverty alleviation. Now, they are working towards the revitalization of the countryside in 2035.
- Line 749-753: no need to repeat the process you have mentioned in the previous sections.
Answer:
Thank you very much for pointing out the problem. We have removed this section. Revised portions are marked in red in the “revised manuscript”.
- Section 5.1: the four points are general conclusions, need to related the data you got from the model to do the further analysis and discussion.
Answer:
Thank you very much for pointing out the problem. We have revised this section. Revised portions are marked in red in the “revised manuscript”.
Besides, we can learn from this case. First, insist on the participation of the residents. Respect the residents, mobilize them, rely on them, and enhance the endogenous motivation of the rural masses to develop the countryside and achieve revitalization. The Government will also strengthen the internal motivation of rural people to develop the countryside and achieve revitalization. The establishment of an internal mechanism for the peasants to participate in the development of rural tourism on their own, giving them the right to have a full voice in economic activities, and promoting the development model of “small farmers, big industries” in rural economic activities with family management as the carrier, so that the peasants can start their own businesses and realize the prosperity of the people in the process of rural revitalization. Second, adhere to ecological tourism. Green water and green mountains are golden mountains. Taking the protection of the ecological environment of the countryside as the starting point, we will play the characteristic card, strengthen the deep integration of the characteristic town with tourism, ecology, service industry and agriculture, create a characteristic countryside that no one has, no one has, and no one is special, and improve the attractiveness of out-of-town businessmen and tourists. In the process of implementing the rural revitalization strategy and realizing the integrated development of urban and rural areas, the construction of special and strong, clustered and combined, refined and beautiful, living and new special rural tourism resorts. Through the development of eco-tourism, more and more people in Steep Shahe Village have been able to eat "tourism rice" and embark on the road to prosperity, which has greatly boosted the regional economic development and poverty alleviation, and laid a solid foundation for the implementation of the rural revitalization strategy.
- Section 5.2: the tripartite relationship is not a novel concern in recent studies, there are many references discuss about it. Relating the poverty areas to ecotourism is an interesting idea, however, should consider the potential for implement.
Answer:
Thank you very much for pointing out the problem. We have revised this section. Revised portions are marked in red in the “revised manuscript”.
This research extends the current knowledge on the rural tourism industry with some important research dimensions. Most of the previous studies considered the rural tourism industry to be a simple view. The relevant studies on the problems and countermeasures in rural tourism [12, 15, 16], and study the rural tourism’ pathways and methods [55-62]. This study introduces the tripartite evolutionary game model and focuses on the tourism enterprises, local residents and government on rural ecotourism in terms of evolutionary game. Then, we use simulation method and case study to analyze the stability of interactions among the stakeholders and determine an equilibrium solution in the finite rationality case.
Some previous studies especially emphasized the evolutionary stable strategies of local governments, tourism enterprises and residents in the development and construction of ecotourism in ecologically fragile areas [32], and ignored the rural ecotourism among the three stakeholders in the poverty-stricken areas of China. In this research, we analyze how the different strategies of the three parties affect the rural ecotourism project and implications for rural revitalization. Based on the findings of the study, this paper provides countermeasures and recommendations for tourism enterprises, government, and local residents.
Although this study has significant theoretical and practical implications, it also has several limitations that could be explored in future research. The first is that the parameters of the evolutionary game payment matrix do not take into account all the factors affecting rural ecotourism. And the second is that some of the parameters of the payment matrix may not be precisely assigned due to the constraints of the survey respondents and conditions, which is where further research should be conducted in the future. In addition, other players are important subjects that should be considered when establishing the evolutionary game model. Future research may investigate more stakeholders or consider the technology innovation, innovation ecosystem with the poverty areas to ecotourism.
- Section 5.3: As previous comments, the ideas are general, the practical suggestions should come from the real data and analysis with detailed description.
Answer:
Thank you very much for pointing out the problem. We have revised this section. Revised portions are marked in red in the “revised manuscript”.
(1) For the government, it is necessary to increase the publicity and promo-tion of relevant rural ecotourism policies, so as to enhance local residents' awareness of the relevant policies; increase the incentives for local residents of active participation; increase the incentives for enterprises that substantially implement rural ecotourism and the penalties for enterprises that falsely implement rural ecotourism; and improve the monitoring system to dynamically monitor the implementation of rural ecotourism projects by enterprises. It will increase the likelihood of fraudulent rural ecotourism projects being detected by tourism companies. The government should place rural revitalization in a prominent position, take the construction of a rural revitalization demonstration belt as an important grasp, plan carefully, highlight the characteristics according to local conditions, make full use of the linkage and radiation of the surrounding characteristic towns, and build a rural revitalization demonstration village with high standards. In addition, the local government should establish a fast and smooth information network, adopt modern information means such as e-commerce, negotiate projects through the Internet and other forms, and strive to expand online investment, while implementing entrusted investment or intermediary investment.
(2) For tourism enterprises, the pursuit of profit maximize is their goal. However, rural ecotourism projects are heavily invested, and because returns are difficult and slow to achieve in the short term, tourism enterprises may be reluctant to undertake substantive implementation of rural ecotourism projects. If tourism enterprises falsely implement rural ecotourism projects, they will suffer a loss of overall social benefits. Therefore, tourism enterprises should also take social responsibility, have a long-term vision and respond to the government’s call to implement rural ecotourism projects for local residents, so as to better serve rural revitalization.
(3) For local residents, they should change their traditional concepts, actively learn about the relevant rural ecotourism policies, and actively participate in rural ecotourism projects with the help of the government and tourism enterprises, so as to successfully realize their own poverty alleviation and re-employment, improve their own skills and meet the needs of rural revitalization. At the same time, based on the actual demand for the development and operation of rural eco-tourism resources, local residents are provided with skills training in the dissemination of rural eco-tourism cultural resources, the transmission of special skills and the operation of B&Bs, so as to expand the audience for vocational education and strengthen the effectiveness of vocational education.

Reviewer 2 Report
The manuscript prepared for review concerns the regime of rural ecotourism stakeholders in poverty-stricken areas of China: implications for rural revitalization. The topic is interesting, but the manuscript contains numerous flaws that need to be corrected. The manuscript requires editorial corrections. It is not entirely clear whether the Introduction chapter deals with rural ecotourism in general or in a specific one - China. The authors do not specify where in the chapter they write in general and where in detail. The Literature Review chapter is a continuation of the Introduction chapter. Thus, in the Introduction, which should be a short and concise chapter, Authors should present the background of their research and elaborate on it in the Literature Review. The specificity of rural ecotourism in China is too poorly described - it applies to the entire manuscript. Do the vines take into account the specificity of rural ecotourism in China? There is no clear-cut research goal. It is not clear from the manuscript whether the authors describe China in a given place or refer to general truths. The manuscript is completely devoid of discussion, which is unacceptable in the case of scientific articles.
Author Response
To Reviewer
Dear Reviewer,
Thank you for your letter and for the reviewer’ comments concerning our manuscript entitled “The regime of rural ecotourism stakeholders in poverty-stricken areas of China: implications for rural revitalization”(ID: ijerph-1323897).
Those comments are all valuable and helpful for revising and improving our paper, as well as the important guiding significance to our researches. We have studied comments carefully and have made correction which we hope meet with approval. Revised portions are marked by track changes in the “revised manuscript”. The main corrections in the paper and the responds to the reviewer’s comments are as follows.
Responds to the reviewer’s comments 2:
- The manuscript prepared for review concerns the regime of rural ecotourism stakeholders in poverty-stricken areas of China: implications for rural revitalization. The topic is interesting, but the manuscript contains numerous flaws that need to be corrected. The manuscript requires editorial corrections. It is not entirely clear whether the Introduction chapter deals with rural ecotourism in general or in a specific one - China. The authors do not specify where in the chapter they write in general and where in detail. The Literature Review chapter is a continuation of the Introduction chapter. Thus, in the Introduction, which should be a short and concise chapter, Authors should present the background of their research and elaborate on it in the Literature Review. The specificity of rural ecotourism in China is too poorly described - it applies to the entire manuscript. Do the vines take into account the specificity of rural ecotourism in China? It is not clear from the manuscript whether the authors describe China in a given place or refer to general truths.
Answer:
Thank you very much for pointing out the problem. We have revised our introduction and literature review. We have added the specificity of rural ecotourism in China. Revised portions are marked in red in the “revised manuscript”. As following:
- 1. Introduction
Rural tourism has strong market advantages, powerful blood-making functions, emerging industrial vitality and huge driving effects [1-2]. On the one hand, it is essential to promote harmony and achieving international exchanges; on the other hand, it can drive the economic development of poor regions, increasing employment opportunities for the poor, and play an increasingly important role in the cause of poverty reduction in the world [3-4]. In the past, rural tourism has also played a huge role in China’s poverty alleviation and development, with remarkable results [5-8]: firstly, with the implementation and upgrading of the action plan for the transformation and upgrading of tourism infrastructure in the “three regions and three states” deep poverty areas and the “three regions and three states” tourism ring road, the tourism ring road has been implemented. Secondly, with the implementation of the action plan for upgrading tourism infrastructure in the “three regions and three states” and thetourism loop, rural tourism in the “three regions and three states” deep poverty areas has achieved significant results in poverty alleviation. On 18 May 2020, the Information Office of the State Council reported that as of 17 May, 780 counties had been declared out of poverty, leaving only 52 poor counties in seven provinces and regions [8-9]. The poor counties are a major achievement in the battle against poverty, and from the way the poor counties have been removed from poverty and its effectiveness, there is a lot of “tourism power” in this. Finally, in recent years, rural tourism has explored and developed practical and effective ways to help alleviate poverty, such as, Manaus in Brazil, Cancun Island in Mexico, Las Vegas in the United States, the Baleares Islands in Spain, the French Alps, Phuket in Thailand, Bali in Indonesia, the Sinai Peninsula in Egypt and Morocco in Africa, which were previously very backward places, have all become world-famous tourist destinations as a result of rural tourism development, with great social, economic and cultural improvement and development. In particular, as highlighted by other authors [10-11], the reinforcement of local food productions and short food supply chains might play a key role in rural ecotourism revitalization. In China, such as “scenic spots leading villages”, “capable people leading households”, “cooperatives+ farmers”, and “enterprises+ farmers”, which are important achievements in China’s poverty alleviation efforts. Although rural tourism has made a series of achievements and will enter the final year of the battle against poverty in 2020, there are still many problems that need to be solved in the next “rural revitalization era”. For example, many regions lack planning in rural tourism development, blindly develop tourism projects without in-depth study of local tourism resources and advantages [12-13], design similar tourism products without characteristics [14-15], excessively pursue short-term economy without long-term planning [11,15,16-17], and establish “temporary rural tourism projects”. In some regions, in the process of rural tourism, there is an imbalance between the interests of the poor, the local government, tourism developers and tourists [12, 18]. In other areas, there are poor infrastructure facilities and a lack of professional management personnel, which makes it difficult to attract tourists [19-20]. Therefore, to face these problems, the Chinese government needs to seize the policy opportunities, build on the existing foundation and experience, further improve the work of rural tourism to help alleviate poverty, consolidate the achievements of rural tourism, tell the story of rural tourism in China, and further advance towards the “rural revitalization in 2035”.
The development of rural ecotourism in poverty-stricken areas has become a key to achieving environmental protection, rural revitalization and sustainable development [10-11,21-23]. Rural ecotourism is an emerging type of tourism which gained attention in rural areas rich in ecological resources for the purpose of promoting the development of rural areas and protecting the rural ecological environment [24-26]. In the context of ecological civilization, the development of green rural tourism has considerable strategic importance, not only to improve the ecological environment but also to increase economic income for local residents, creating a beautiful and livable environment and promoting common prosperity [27]. In recent years, rural ecotourism has become more and more recognized by the public as a new mode of tourism, and the development of rural tourism has become an effective way to promote rural revitalization [4]. Compared with sightseeing spots and city tours, rural ecotourism uses the unique resources of the countryside to change tourism from passive visits to active participation, making people feel the simplicity and happiness of returning to the basics, which is increasingly recognized by the public [28].
However, there are problems in the development of rural ecotourism, such as homogenization of projects, sloppy management, difficulties in financing, shortage of talents and serious environmental pollution [29]. Due to the asymmetrical distribution of rural ecotourism resources, uneven economic development between urban and rural areas, towns and villages, villagers and local governments often develop “rural ecotourism” under the banner of ecological conservation to pursue their own interests and performance, while secretly trampling on the natural ecological environment of the countryside [30-32].There is no balance between pre-conservation development and post-conservation development, which may lead to further deterioration of the rural ecological environment [33-35]. In the process of developing and building rural ecotourism, tourism enterprises will adjust their strategies according to local government policies, but may sacrifice rural ecological environment and engage in false promotion of rural ecotourism in order to maximize short-term benefits [36-38]. The strategies of tourism enterprises largely determine the implementation strategies of other stakeholders [39-40]. However, existing research has surprisingly failed to focus on stakeholder conflicts of interest and whether stakeholders are authentic in implementing rural ecotourism, especially in relatively poor areas. Whether existing models of cooperation and incentive policies are feasible, how well the mechanisms work and what impact they have needs to be more fully demonstrated. Therefore, it is necessary to study the stakeholders’ in interaction and strategic choice in the development and construction of rural ecotourism in poverty-stricken areas [41-43]. In our study, we plan an ecotourism system and analysis the relationship of the tourism enterprises, local residents and government by using an evolutionary game theory. Based on the theoretical analysis, an evolution game model for the three stakeholders is developed and the evolution process of strategies is described by replicator dynamic equations [32, 44]. Then, we use simulation method and case study to analyze the stability of interactions among the stakeholders and determine an equilibrium solution in the finite rationality case. Finally, specific control strategies are proposed to suppress instability and an ideal evolutionarily stable strategy is obtained. This provides a theoretical basis for achieving a win-win situation among the three parties. Our paper has two contributions that are not fully addressed in the previous literature. Firstly, we focus on the tourism enterprises, local residents and government on rural ecotourism in terms of evolutionary game, and analyze the regime mechanism of rural ecotourism among the three stakeholders in the poverty-stricken areas of China. Secondly, we use case study and simulation to analyze how the different strategies of the three parties affect the rural ecotourism project and implication for rural revitalization. Based on the findings of the study, this paper provides countermeasures and recommendations for tourism enterprises, government, and local residents.
- Literature Review
Rural Tourism has long been recognized in certain parts of Europe as an effective catalyst of rural socio-economic regeneration for over a hundred years [45]. Along with the development of rural tourism worldwide, rural tourism concept has many interpretations. Tourism activity in rural areas has remarkably increased in all the developed countries and developing countries worldwide, which has played a key role in the development of rural areas that were economically and socially depressed [46-47]. Rural tourism is a multi-stakeholder process that requires the joint efforts of all stakeholders [30, 32, 36, 48-49]. The relevant studies on rural tourism include the following aspects. Firstly, study on the problems and countermeasures in rural tourism. In view of the problems in rural tourism, such as insufficient industrial characteristics, poor quality of residents and poor channels of return of benefits, there is a need to precisely characterise the rural tourism industry, invest in human resources for rural tourism and carefully design the return mechanism of tourism benefits [12, 15, 16]. In the process of rural tourism, problems such as the ‘tragedy of the commons’ can occur, which can hinder the development of rural tourism projects and require specific analysis [50]. In China, the following problems exist in rural tourism: weakness of high value-added industrial links, lack of impetus from core rural tourism enterprises, insufficient cooperation in rural tourism projects, single function of rural tourism industry, short industrial chains and insufficient localization of rural industrial chains [6, 51]. Therefore, we should actively cultivate core rural tourism enterprises [52], strengthen the integration of the rural tourism industry chain [53], accelerate the localization of the rural tourism industry chain and strengthen regional cooperation in rural tourism [53-54]. The second is the study of rural tourism models and pathways. It is necessary to further explore the three major types of rural tourism models: top-down, bottom-up and top-down cooperation, and to propose rural tourism models [55-56], including the government-enterprise cooperation model, the strategic alliance model, the leisure agriculture and rural tourism model and the regional linkage model [57], the rural ecological agriculture model, the tourism+ featured town model, the O-RHB (Organization-Resource-Humanity-Benefit) model and the collaborative participation model of multiple subjects [58-59]. Thirdly, it is a study on the mechanism and effect of rural tourism. With the help of methods such as the Moran index and the spatial Durbin model, explore the spatial correlation characteristics and spatial spillover effects between the level of tourism development and poverty alleviation in each dimension from a multi-dimensional perspective, and conclude that tourism development has a significant effect on poverty alleviation in the economic, living and environmental dimensions [59]. Alternatively, the number of people living in poverty was used as a direct poverty measure, and a social accounting matrix was used to simulate the poverty alleviation effect of tourism, which was found to be an industrial tool for poverty alleviation in Ecuador [60]. Some scholars outside of the first instance have used a three-stage DEA (Data Envelopment Analysis) model to analyze the efficiency of rural tourism in different regions, thus providing valuable strategies and suggestions to promote the smooth implementation of tourism poverty alleviation [61-62]. Of course there are some who argue that rural tourism does not always bring positive poverty alleviation effects, but may also bring negative effects such as environmental changes and loss of traditional culture [63].
The term ecotourism is surrounded by confusion [64]. It has been defined as ‘‘responsible travel to natural areas that conserves the environment and sustains the well-being of local people [65]’’. However, it is contended here that, regardless of definition, ecotourism is an instigator of change. It is inevitable that the introduction of tourists to areas that were previously seldom visited by outsiders will place new demands upon the environment associated with new actors, new activities, and new facilities. This will involve the forging of new relationships between people and environment, between peoples with different life-styles, and between a wide variety of forces for both change and stability. These forces act at a diversity of scales from global to local. Change is desired by most of the players involved in ecotourism, many of whom would like to see what they regard as an improvement in the existing situation. Rural ecotourism is a kind of tourism that takes the protection of the natural ecological environment as the premise and relies on the good natural ecological environment and unique human resources in the countryside to carry out ecological experience, ecological education, ecological cognition, while obtaining physical and mental enjoyment [66]. Rural ecotourism is the main form of sustainable tourism [21, 67]. Rural ecotourism takes the countryside as the backdrop and the distinctive characteristic environment as the main landscape [68]. It mainly refers to the tourism model of ecological education, ecological experience and ecological awareness based on the concept of sustainable development, ecological environmental protection, harmonious development of human and nature, and the goal of maintaining the coordinated economic, social and environmental development of rural areas, relying on a better natural ecological environment and unique human ecological system [27, 69]. Currently, environmental pollution and damage in rural tourism areas are serious and all stakeholders are deeply affected by the environmental damage in rural tourism areas [31, 70]. In order to develop sustainable rural tourism and promote rural ecological and environmental protection and economic development, the relationships and evolutionary processes of the main stakeholders should be mainly analyzed and studied [71-73]. As a method tool, evolutionary game theory has been applied to many fields [74-79]. In recent years, evolutionary game theory has been increasingly applied in the field of rural tourism [80-81] or ecotourism [32, 82], with less attention paid to the poverty-stricken areas, especially in China. Previous studies have mainly considered the relationship between government, firms or residents and firms. However, few scholars have examined the rural ecotourism stakeholders in poverty-stricken areas, highlighting implications for rural revitalization. This paper constructs a tripartite evolutionary game model from the perspective of an evolutionary game combined with simulation and case study analysis to address the issues of local government regulation and resident participation in the implementation of rural ecotourism by tourism enterprises in the construction of rural ecotourism projects. The model is used to analyze the strategy combinations of the three stakeholders, and to provide a long-term stable implementation strategy for the three stakeholders in the rural ecotourism project.
Steep Shahe Village, Huoshan County, Anhui Province, is located in the hinterland of Dabie Mountain, with the Dabie Mountain National Scenic Byway and the most beautiful tourism loop of Huoshan passing through the territory. The village is surrounded by mountains and beautiful scenery. The village has a total area of 1,600 square kilometres, including 12,564 acres of mountains and forests, 1,683 acres of arable land, and is rich in resources such as tea, Chinese herbs and silkworms. In the past, it was a typical mountainous village with poor economic development due to the high mountains and dense forests and closed traffic. In the past, it was a typical mountainous village with poor economic development due to the high mountains and dense forests and closed traffic. The Huoshan County government introduced Jiangsu Huaqiang Group, which invested 465 million USD to develop Steep Shahe Village in 2016, putting the village on the fast track to poverty alleviation through tourism, forming a high-end ecological tourism resort and hot spring town with distinctive self characteristics, integrating tourism, leisure, holiday, entertainment, sports, health and retirement. At the same time, through the development of high quality organic agriculture and the use of financial means to promote consumption, the village has been able to achieve better poverty alleviation results in the short term, allowing this national rural tourism poverty alleviation focus village to regain momentum.
From the above analysis: The different initial game states lead to different evolutionary stability strategy (ESS), with the final choice of which strategy depends on the probability of the various strategies initially chosen. In the long run, an ideal stable state is one in which tourism enterprises substantially undertake rural ecotourism, local residents actively participate, and the government strictly regulates. The government’s promotion of rural ecotourism policies is low, or local residents have a low understanding of the policies, local residents will choose to participate negatively. Local residents may engage in other activities while participating in rural ecotourism projects, such as selling agricultural and sideline products, making special handicrafts, etc. If the income from these activities is low, local residents will choose to actively participate in rural eco-tourism projects; if the income from these activities is high, local residents will choose to negatively participate in rural ecotourism projects. When the probability of a tourism enterprise engaging in false rural ecotourism is low and the government discovers this behavior, the tourism enterprise will choose the fake strategy and the government will choose the formal regulation; as the probability of discovery increases, the tourism enterprise will choose the substantial rural ecotourism strategy and the government will choose the strict regulation strategy. In this way, through the efforts of the Huoshan County Government, Huaqiang and the residents of Steep Shahe, Steep Shahe Village was transformed from a poor village to a high-end ecotourism resort. Since its opening in October 2017, Steep Shahe Hot Spring Town has received a total of more than one million visitors and achieved a comprehensive income of 77.56 million USD. Family hotels and farmhouse restaurants have blossomed in all aspects around Steep Shahe Village and the Shangtu City market town. In 2017 and 2018, 103 new farmhouses and farmhouse inns were added to Steep Shahe Village, and farm specialties such as tea leaves, local eggs, black-haired pigs, red lantern peppers and Steep Shahe vermicelli are very popular. The hot spring town has also achieved direct employment for farmers, with more than 500 migrant workers in the town working in the hot spring town, driving 86 households and 230 people in poverty to achieve stable employment, with an average household income of more than 4653.54 USD. The hot spring town and related projects have transferred a total of 1,737 acres of land to 461 households, with annual transfer rents totalling 168148.04 USD, directly increasing the income of farmers and ensuring that they are not unemployed and have guaranteed income. By combining the role of the organization with the role of the market, and the cooperative helping to sell with the farmers' own sales, 50,000 kg of organic vermicelli have been sold by the cooperative since 2019, including 15,000 kg for 22 poor households, driving 180 households, including 45 poor households, with an average household income of 775.59 USD, and an increase of 31023.62 USD in the village collective economy. During the epidemic, Steep Shahe Village helped 11 poor households sell 4515 kg of sweet potato vermicelli on their behalf, earning 21010.75 USD, directly increasing the income of poor households and strongly contributing to poverty alleviation. Now, they are working towards the revitalization of the countryside in 2035.
Besides, we can learn from this Chinese case. First, insist on the participation of the residents. Respect the residents, mobilize them, rely on them, and enhance the endogenous motivation of the rural masses to develop the countryside and achieve revitalization. The Government will also strengthen the internal motivation of rural people to develop the countryside and achieve revitalization. The establishment of an internal mechanism for the peasants to participate in the development of rural tourism on their own, giving them the right to have a full voice in economic activities, and promoting the development model of “small farmers, big industries” in rural economic activities with family management as the carrier, so that the peasants can start their own businesses and realize the prosperity of the people in the process of rural revitalization. Second, adhere to ecological tourism. Green water and green mountains are golden mountains. Taking the protection of the ecological environment of the countryside as the starting point, we will play the characteristic card, strengthen the deep integration of the characteristic town with tourism, ecology, service industry and agriculture, create a characteristic countryside that no one has, no one has, and no one is special, and improve the attractiveness of out-of-town businessmen and tourists. In the process of implementing the rural revitalization strategy and realizing the integrated development of urban and rural areas, the construction of special and strong, clustered and combined, refined and beautiful, living and new special rural tourism resorts. Through the development of eco-tourism, more and more people in Steep Shahe Village have been able to eat "tourism rice" and embark on the road to prosperity, which has greatly boosted the regional economic development and poverty alleviation, and laid a solid foundation for the implementation of the rural revitalization strategy.

Reviewer 3 Report
Revision of the manuscript:
The regime of rural ecotourism stakeholders in poverty-stricken areas of China: implications for rural revitalization.
Authors: Xia Cao, Keke Sun*, Zeyu Xing, and Weijia Li.
In general the article is interesting and the novelty is adequate. The article might be of technical and practical interest to many readers of this journal. For these reasons, I believe that this paper could be accepted for publication, but only after Minor Revisions
Here following the corrections that need to be made:
Title:
Ok
Abstract:
Line 10: remove when studying rural ecotourism the . just start with Rural ecotourism…
Line 10: replace better understood with defined
Line 16: Then, a simulation…. case was used.
Line 18: replace are with were
Line 19: replace is with was
Introduction:
Line 28: it is essential to promote…
Line 29: exchanges; on the other hand,…
Line 30: increasing
Line 33 – 41: please rephrase and reduce to the minimum the repetition of “three regions and three states”. You repeated too many times.
Line 45- 49: in my opinion this is essential to increase local food productions and local tourism. However you did not mention any example outside from China. So I strongly suggest to add one of two sentences to strengthen this concept citing similar scenarios in all over the world (like, for example, the rediscover of ancient wheat to valorize regional scenarios).
You could add a couple of sentences like this:
..alleviation efforts. In particular, as highlighted by other authors [10, 11], the reinforcement of local food productions and short food supply chains might play a key role in rural ecotourism revitalization.
Please add these paper in the references list and scale of two position the old references in the text and in the references list.
[10] Cappelli, A., & Cini, E. (2021). Challenges and Opportunities in Wheat Flour, Pasta, Bread, and Bakery Product Production Chains: A Systematic Review of Innovations and Improvement Strategies to Increase Sustainability, Productivity, and Product Quality. Sustainability, 13(5), 2608.
[12] Cappelli, A., Lupori, L., & Cini, E. (2021). Baking technology: A systematic review of machines and plants and their effect on final products, including improvement strategies. Trends in Food Science & Technology, 115, 275–284.
Line 60: therefore, to face these problems, …
Line 65-67: please cite again [10, 11] here.
Line 67: Rural ecotourism is an emerging type of tourism which gained attention in rural….
Line 72: creating….. promoting…
Line 86: missing a space here
Line 98: remove in
2 literature review:
Line 155-156: Rural ecotourism takes the countryside …
Line 173: areas, highlighting implications for rural revitalization
3 model building and analysis:
Line 251 : remove some
Line 425, 427, and 428: remove double space here
Line 467: missing a space between of and M
Line 481: this must be number 4 NOT 3 .
Please correct as 4. Case Analysis ….
Line 701 – 746: in my opinion here you should emphasize possible solutions in the case of significant fall in tourism due to covid-19 pandemic or from other critical issue. I mean that you should suggest alternative solutions for local enterprises which should base their business and income in selling their food products in alternative ways in the case of tourism restriction (increase local knowledge of the products so that the products can be "absorbed" (purchased) locally; internet selling; government compensation and funding etc..)
5 Conclusions
Line 794: for the government, it is necessary to increase…
Line 818 822: what about policies aimed to increase educational school education ?
Author Response
To Reviewer
Dear Reviewer,
Thank you for your letter and for the reviewer’ comments concerning our manuscript entitled “The regime of rural ecotourism stakeholders in poverty-stricken areas of China: implications for rural revitalization”(ID: ijerph-1323897).
Those comments are all valuable and helpful for revising and improving our paper, as well as the important guiding significance to our researches. We have studied comments carefully and have made correction which we hope meet with approval. Revised portions are marked by track changes in the “revised manuscript”. The main corrections in the paper and the responds to the reviewer’s comments are as follows.
Responds to the reviewer’s comments 3:
In general the article is interesting and the novelty is adequate. The article might be of technical and practical interest to many readers of this journal. For these reasons, I believe that this paper could be accepted for publication, but only after Minor Revisions
- Abstract: Line 10: remove when studying rural ecotourism the . just start with Rural ecotourism…Line 10: replace better understood with defined
Answer:
Thank you very much for pointing out the problem. We have revised our abstract. Revised portions are marked in red in the “revised manuscript”. As following:
The rural ecotourism system can be defined as a complex association of stakeholders.
- Line 16: Then, a simulation…. case was used.
Answer:
Thank you very much for pointing out the problem. We have revised our abstract. Revised portions are marked in red in the “revised manuscript”. As following:
Then, a simulation method and case was used to analyze the stability of interactions among the stakeholders and determine an equilibrium solution in the finite rationality case.
- Line 18: replace are with were
Answer:
Thank you very much for pointing out the problem. We have revised our abstract. Revised portions are marked in red in the “revised manuscript”. As following:
Finally, specific control strategies were proposed to suppress instability and an ideal evolutionarily stable strategy was obtained.
- Line 19: replace is with was
Answer:
Thank you very much for pointing out the problem. We have revised our abstract. Revised portions are marked in red in the “revised manuscript”. As following:
Finally, specific control strategies were proposed to suppress instability and an ideal evolutionarily stable strategy was obtained.
- Introduction: Line 28: it is essential to promote…
Answer:
Thank you very much for pointing out the problem. We have revised our introduction. Revised portions are marked in red in the “revised manuscript”. As following:
On the one hand, it is essential to promote harmony and achieving international exchanges;
- Line 29: exchanges; on the other hand,…
Answer:
Thank you very much for pointing out the problem. We have revised our introduction. Revised portions are marked in red in the “revised manuscript”. As following:
On the one hand, it is essential to promote harmony and achieving international exchanges; on the other hand, it can drive the economic development of poor regions,
- Line 30: increasing
Answer:
Thank you very much for pointing out the problem. We have revised our introduction. Revised portions are marked in red in the “revised manuscript”. As following:
Increasing employment opportunities for the poor,
- Line 33 -41: please rephrase and reduce to the minimum the repetition of “three regions and three states”. You repeated too many times.
Answer:
Thank you very much for pointing out the problem. We have revised our introduction. We have removed the redundant of “three regions and three states”. Revised portions are marked in red in the “revised manuscript”. As following:
The tourism ring road has been implemented. Secondly, with the implementation of the action plan for upgrading tourism infrastructure in the “three regions and three states” and the tourism loop, rural tourism in the “three regions and three states” deep poverty areas has achieved significant results in poverty alleviation.
- Line 45- 49: in my opinion this is essential to increase local food productions and local tourism. However you did not mention any example outside from China. So I strongly suggest to add one of two sentences to strengthen this concept citing similar scenarios in all over the world (like, for example, the rediscover of ancient wheat to valorize regional scenarios).
Answer:
Thank you very much for pointing out the problem. We have revised this section, and added some examples outside from China. Revised portions are marked in red in the “revised manuscript”. As following:
Finally, in recent years, rural tourism has explored and developed practical and effective ways to help alleviate poverty, such as, Manaus in Brazil, Cancun Island in Mexico, Las Vegas in the United States, the Baleares Islands in Spain, the French Alps, Phuket in Thailand, Bali in Indonesia, the Sinai Peninsula in Egypt and Morocco in Africa, which were previously very backward places, have all become world-famous tourist destinations as a result of rural tourism development, with great social, economic and cultural improvement and development.
- You could add a couple of sentences like this: ..alleviation efforts. In particular, as highlighted by other authors [10, 11], the reinforcement of local food productions and short food supply chains might play a key role in rural ecotourism revitalization. Please add these paper in the references list and scale of two position the old references in the text and in the references list. [10] Cappelli, A., & Cini, E. (2021). Challenges and Opportunities in Wheat Flour, Pasta, Bread, and Bakery Product Production Chains: A Systematic Review of Innovations and Improvement Strategies to Increase Sustainability, Productivity, and Product Quality. Sustainability, 13(5), 2608. [12] Cappelli, A., Lupori, L., & Cini, E. (2021). Baking technology: A systematic review of machines and plants and their effect on final products, including improvement strategies. Trends in Food Science & Technology, 115, 275–284.
Answer:
Thank you very much for pointing out the problem. We have revised this section, and added the references in our list. Revised portions are marked in red in the “revised manuscript”. As following:
In particular, as highlighted by other authors [10-11], the reinforcement of local food productions and short food supply chains might play a key role in rural ecotourism revitalization.
The development of rural ecotourism in poverty-stricken areas has become a key to achieving environmental protection, rural revitalization and sustainable development [10-11,21-23].
[10] Cappelli, A., & Cini, E. (2021). Challenges and Opportunities in Wheat Flour, Pasta, Bread, and Bakery Product Production Chains: A Systematic Review of Innovations and Improvement Strategies to Increase Sustainability, Productivity, and Product Quality. Sustainability, 13(5), 2608.
[11] Cappelli, A., Lupori, L., & Cini, E. (2021). Baking technology: A systematic review of machines and plants and their effect on final products, including improvement strategies. Trends in Food Science & Technology, 115, 275–284.
- Line 60: therefore, to face these problems, …
Answer:
Thank you very much for pointing out the problem. We have revised this sentence. Revised portions are marked in red in the “revised manuscript”. As following:
Therefore, to face these problems, the Chinese government needs to seize the policy opportunities,
- Line 67: Rural ecotourism is an emerging type of tourism which gained attention in rural….
Answer:
Thank you very much for pointing out the problem. We have revised this section. Revised portions are marked in red in the “revised manuscript”. As following:
Rural ecotourism is an emerging type of tourism which gained attention in rural areas rich in ecological resources for the purpose of promoting the development of rural areas and protecting the rural ecological environment.
- Line 72: creating….. promoting…
Answer:
Thank you very much for pointing out the problem. We have revised this sentence. Revised portions are marked in red in the “revised manuscript”. As following:
Creating a beautiful and livable environment and promoting common prosperity.
- Line 98: remove in
Answer:
Thank you very much for pointing out the problem. We have revised this section. Revised portions are marked in red in the “revised manuscript”.
- Literature review: Line 155-156: Rural ecotourism takes the countryside …
Answer:
Thank you very much for pointing out the problem. We have revised this sentence. Revised portions are marked in red in the “revised manuscript”. As following:
Rural ecotourism takes the countryside as the backdrop and the distinctive characteristic environment as the main landscape.
- Line 173: areas, highlighting implications for rural revitalization
Answer:
Thank you very much for pointing out the problem. We have revised this sentence. Revised portions are marked in red in the “revised manuscript”. As following:
However, few scholars have examined the rural ecotourism stakeholders in poverty-stricken areas, highlighting implications for rural revitalization.
- Model building and analysis: Line 251 : remove some
Answer:
Thank you very much for pointing out the problem. We have revised this sentence. Revised portions are marked in red in the “revised manuscript”. As following:
And earn income, which is denoted by.
- Line 425, 427, and 428: remove double space here
Answer:
Thank you very much for pointing out the problem. We have revised this section. Revised portions are marked in red in the “revised manuscript”.
- Line 467: missing a space between of and M
Answer:
Thank you very much for pointing out the problem. We have revised this section. Revised portions are marked in red in the “revised manuscript”.
- Line 481: this must be number 4 NOT 3 . Please correct as 4. Case Analysis ….
Answer:
Thank you very much for pointing out the problem. We have revised this section. Revised portions are marked in red in the “revised manuscript”. As following:
- Case Analysis and Numerical Simulation
- Line 701 – 746: in my opinion here you should emphasize possible solutions in the case of significant fall in tourism due to covid-19 pandemic or from other critical issue. I mean that you should suggest alternative solutions for local enterprises which should base their business and income in selling their food products in alternative ways in the case of tourism restriction (increase local knowledge of the products so that the products can be "absorbed" (purchased) locally; internet selling; government compensation and funding etc..)
Answer:
Thank you very much for pointing out the problem. We have revised this section. Revised portions are marked in red in the “revised manuscript”. As following:
From the above analysis: The different initial game states lead to different evolutionary stability strategy (ESS), with the final choice of which strategy depends on the probability of the various strategies initially chosen. In the long run, an ideal stable state is one in which tourism enterprises substantially undertake rural ecotourism, local residents actively participate, and the government strictly regulates. The government’s promotion of rural ecotourism policies is low, or local residents have a low understanding of the policies, local residents will choose to participate negatively. Local residents may engage in other activities while participating in rural ecotourism projects, such as selling agricultural and sideline products, making special handicrafts, etc. If the income from these activities is low, local residents will choose to actively participate in rural eco-tourism projects; if the income from these activities is high, local residents will choose to negatively participate in rural ecotourism projects. When the probability of a tourism enterprise engaging in false rural ecotourism is low and the government discovers this behavior, the tourism enterprise will choose the fake strategy and the government will choose the formal regulation; as the probability of discovery increases, the tourism enterprise will choose the substantial rural ecotourism strategy and the government will choose the strict regulation strategy. In this way, through the efforts of the Huoshan County Government, Huaqiang and the residents of Steep Shahe, Steep Shahe Village was transformed from a poor village to a high-end ecotourism resort. Since its opening in October 2017, Steep Shahe Hot Spring Town has received a total of more than one million visitors and achieved a comprehensive income of 77.56 million USD. Family hotels and farmhouse restaurants have blossomed in all aspects around Steep Shahe Village and the Shangtu City market town. In 2017 and 2018, 103 new farmhouses and farmhouse inns were added to Steep Shahe Village, and farm specialties such as tea leaves, local eggs, black-haired pigs, red lantern peppers and Steep Shahe vermicelli are very popular. The hot spring town has also achieved direct employment for farmers, with more than 500 migrant workers in the town working in the hot spring town, driving 86 households and 230 people in poverty to achieve stable employment, with an average household income of more than 4653.54 USD. The hot spring town and related projects have transferred a total of 1,737 acres of land to 461 households, with annual transfer rents totalling 168148.04 USD, directly increasing the income of farmers and ensuring that they are not unemployed and have guaranteed income. By combining the role of the organization with the role of the market, and the cooperative helping to sell with the farmers' own sales, 50,000 kg of organic vermicelli have been sold by the cooperative since 2019, including 15,000 kg for 22 poor households, driving 180 households, including 45 poor households, with an average household income of 775.59 USD, and an increase of 31023.62 USD in the village collective economy. During the epidemic, Steep Shahe Village helped 11 poor households sell 4515 kg of sweet potato vermicelli on their behalf, earning 21010.75 USD, directly increasing the income of poor households and strongly contributing to poverty alleviation. Now, they are working towards the revitalization of the countryside in 2035.
- Conclusions: Line 794: for the government, it is necessary to increase…
Answer:
Thank you very much for pointing out the problem. We have revised this sentence. Revised portions are marked in red in the “revised manuscript”. As following:
For the government, it is necessary to increase the publicity and promotion of relevant rural ecotourism policies,
- Line 818 822: what about policies aimed to increase educational school education ?
Answer:
Thank you very much for pointing out the problem. We have revised this section and added the policies of education. Revised portions are marked in red in the “revised manuscript”. As following:
At the same time, based on the actual demand for the development and operation of rural eco-tourism resources, local residents are provided with skills training in the dissemination of rural eco-tourism cultural resources, the transmission of special skills and the operation of B&Bs, so as to expand the audience for vocational education and strengthen the effectiveness of vocational education.

Round 2
Reviewer 1 Report
The present version of the paper is greatly enhanced, the organization and treatment of the subjects is better in the present form. I believe the authors have been a great job here. Several parts of the paper have been reinforced and justified with new literature. All my concerns regarding the paper were attended. Till I thought the the identification of the stakeholders were clearly explained by test, but It's alright for understanding this paper. Therefore, I feel that the manuscript have the quality for publication.
Reviewer 2 Report
The manuscript has been significantly improved by the authors and is suitable for publication in this form.